# SCALE: Self-uncertainty Conditioned Adaptive Looking and Execution for Vision-Language-Action Models

Hyeonbeom Choi [*]  Daechul Ahn [*]  Youhan Lee  Taewook Kang  Seongwon Cho  Jonghyun Choi [†]

## Abstract

Vision-Language-Action (VLA) models have emerged as a promising paradigm for general-purpose robotic control, with test-time scaling (TTS) gaining attention to enhance robustness beyond training. However, existing TTS methods for VLAs require additional training, verifiers, and multiple forward passes, making them impractical for deployment. Moreover, they intervene only at action decoding while keeping visual representations fixed—insufficient under perceptual ambiguity, where reconsidering *how to perceive* is as important as deciding *what to do*. To address these limitations, we propose SCALE, a simple inference strategy that jointly modulates visual perception and action based on 'self-uncertainty', inspired by *uncertainty-driven exploration* in Active Inference theory—requiring no additional training, no verifier, and only a single forward pass. SCALE broadens exploration in both perception and action under high uncertainty, while focusing on exploitation when confident—enabling adaptive execution across varying conditions. Experiments on simulated and real-world benchmarks demonstrate that SCALE improves state-of-the-art VLAs and outperforms existing TTS methods while maintaining single-pass efficiency. Our code is publicly available at `https://github.com/snumprlab/scale`.

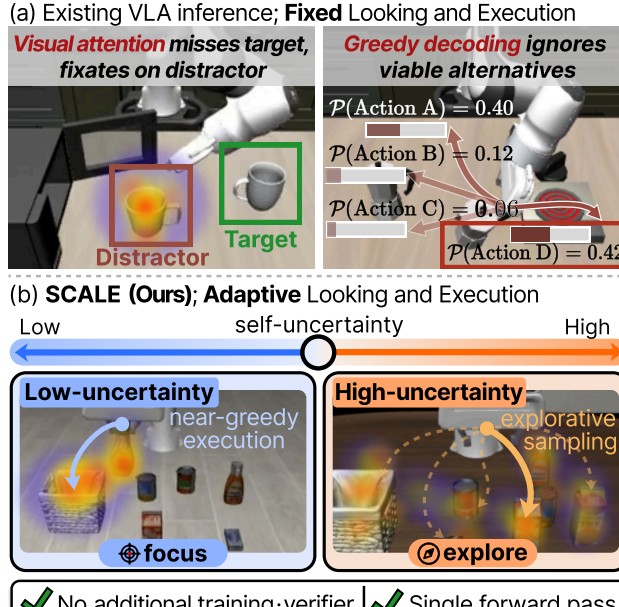

*Figure 1.* Motivation of SCALE. (a) Existing VLA inference relies on a fixed pipeline, where visual attention may miss task-relevant cues (left; red and green boxes) and greedy decoding commits to a single action despite plausible alternatives (right). (b) SCALE addresses these limitations by jointly modulating visual perception and action based on *self-uncertainty*: under low uncertainty, it sharpens attention and performs near-greedy execution; under high uncertainty, it broadens attention and enables explorative sampling.

## 1. Introduction

Vision–Language–Action (VLA) models map multimodal observations and language goals to actions under closed-loop control, offering a promising path toward general-purpose embodied agents (Brohan et al., 2023; Zitkovich et al., 2023; Kim et al., 2024; Pertsch et al., 2025). Among these, *autoregressive* VLAs have emerged as a predominant paradigm: they encode visual observations through a vision encoder and sequentially decode action tokens conditioned on language instructions, seamlessly leveraging pretrained vision–language models (VLMs) with minimal architectural modifications (Zitkovich et al., 2023; Kim et al., 2024; Qu et al., 2025; Driess et al., 2023; Pertsch et al., 2025). Despite their strong generalization capabilities inherited from VLMs, the diversity of real-world environments—where robots encounter novel scenarios that cannot be fully anticipated during training—has driven recent research toward enhancing VLA robustness at *test time*, rather than relying solely on training-time optimization (Kwok et al., 2025;

---

[*]Equal contribution [†]JC is with ECE, IPAI and ASRI in Seoul National University. All authors are with Seoul National University. Correspondence to: Jonghyun Choi <jonghyunchoi@snu.ac.kr>.

*Proceedings of the 43rd International Conference on Machine Learning*, Seoul, South Korea. PMLR 306, 2026. Copyright 2026 by the author(s).

Nakamoto et al., 2024; Yang et al., 2025; Jang et al., 2025).

A prominent strategy for such test-time enhancement is *Test-Time Scaling* (TTS) (Snell et al., 2025), which allocates additional compute at inference to improve performance. Proven effective in LLMs (Snell et al., 2025; Wang et al., 2023) and VLMs (Chen et al., 2024; Zhu et al., 2025), TTS has been recently extended to VLAs via Best-of-$N$ selection (Gao et al., 2023) with external verifier (Kwok et al., 2025; Yang et al., 2025) or self-verification (Jang et al., 2025). However, these approaches have limitations. Practically, they require additional training for verification, degrade under domain shift beyond the verifier's training distribution (Yin et al., 2025; Jang et al., 2025), and incur multiple forward passes that conflict with real-time constraints. Methodologically, existing TTS methods involve only *action decoding* while keeping the visual representation fixed. Yet under *perceptual ambiguity* (*e.g.*, similar distractors), selecting the best action among candidates may be insufficient without reconsidering *how to perceive* the scene (Bajcsy, 1988; Bohg et al., 2017; Xiong et al., 2025; Qu et al., 2025).

To address these limitations, we propose SCALE (**S**elf-uncertainty **C**onditioned **A**daptive **L**ooking and **E**xecution), a simple inference strategy that jointly modulates visual perception and action based on *self-uncertainty*, requiring no additional training or external verifier, and running in a single forward pass. Our approach draws conceptual motivation from *uncertainty-driven exploration* in Active Inference theory (Friston et al., 2016; Schwartenbeck et al., 2019), where agents reduce uncertainty by adapting both perception and action—a principle observed in humans (Daw et al., 2006; Wilson et al., 2014) and formalized in robotics as active perception (Bohg et al., 2017; Bajcsy et al., 2018). This principle naturally raises a question: *how can we quantify self-uncertainty to enable such adaptive modulation?*

Recent work in LLMs has estimated self-uncertainty by measuring how close the predicted output distribution is to uniform, *i.e.*, full ambiguity (Kang et al., 2025). While this captures overall distributional uncertainty, it does not account for the model's decisiveness about its top-1 choice, *i.e.*, how confidently it commits to that selection. In VLAs, this decisiveness is equally important: greedy decoding, used in most VLAs (Kim et al., 2024; Qu et al., 2025), selects the top-1 action for immediate execution—often affecting the environment irreversibly—making confidence in this selection essential for execution reliability. A proper measure of self-uncertainty must therefore capture not only distributional uncertainty but also the model's decisiveness about its top-1 action. To satisfy this requirement, our key idea is to measure where the predicted distribution lies between opposite ends of the certainty spectrum: full certainty (reflecting decisiveness in the top-1 action) and full ambiguity (reflecting overall distributional uncertainty), yielding a

measure that captures both aspects simultaneously.

Specifically, inspired by log-likelihood ratio testing (Neyman & Pearson, 1933; Kullback & Leibler, 1951), which compares two competing hypotheses by measuring their relative likelihood, we formalize this idea by defining two reference distributions—a one-hot distribution centered on the most probable token (full certainty) and a uniform distribution over all tokens (full ambiguity). This yields a bounded, continuous self-uncertainty score computed solely from output logits without additional training (Sec. 3.2). We leverage this to modulate two complementary aspects in VLA (Fig. 1): **(1)** *what to do*, by adjusting action sampling temperature based on token-level uncertainty (Sec. 3.3.1), and **(2)** *how to perceive*, by adjusting visual attention temperature based on step-level uncertainty (Sec. 3.3.2). In closed-loop control, these mechanisms form a feedback loop: uncertainty at one timestep modulates action sampling while simultaneously adjusting visual attention for the next, enabling the model to adapt to varying conditions and execute tasks robustly.

We validate SCALE on simulated and real-world benchmarks across diverse autoregressive VLA architectures, including both seen and unseen scenarios. Our method consistently improves over state-of-the-art (SoTA) VLAs and even outperforms recent TTS VLA approaches that require additional training and multiple inference passes, while maintaining single-pass efficiency suitable for real-time deployment.

## 2. Related Work

**Test-time scaling on VLA models.** Allocating additional compute at inference time has proven effective in LLMs for reasoning and code generation (Snell et al., 2025; Wang et al., 2023), motivating recent extensions to VLAs through generate-and-verify strategies (Nakamoto et al., 2024; Kwok et al., 2025; Yang et al., 2025). V-GPS (Nakamoto et al., 2024) trains an offline RL value function to re-rank sampled actions, and RoboMonkey (Kwok et al., 2025) scales up action verifier training. MG-Select (Jang et al., 2025) avoids external verifiers by using the model's own distribution for self-verification, yet still requires additional training and multiple samples. Despite their effectiveness, these methods share common drawbacks: additional training for verifiers or multiple forward passes, and limited generalization to unseen conditions (Nakamoto et al., 2024; Kwok et al., 2025). In contrast, we propose SCALE that leverages self-uncertainty in the output distribution, enabling adaptive inference in a single forward pass without auxiliary training.

**Uncertainty estimation in generative models.** Quantifying prediction uncertainty has been studied in generative models, particularly LLMs. Early work used output distributions for truncation-based decoding such as top-$k$ (Fan et al.,

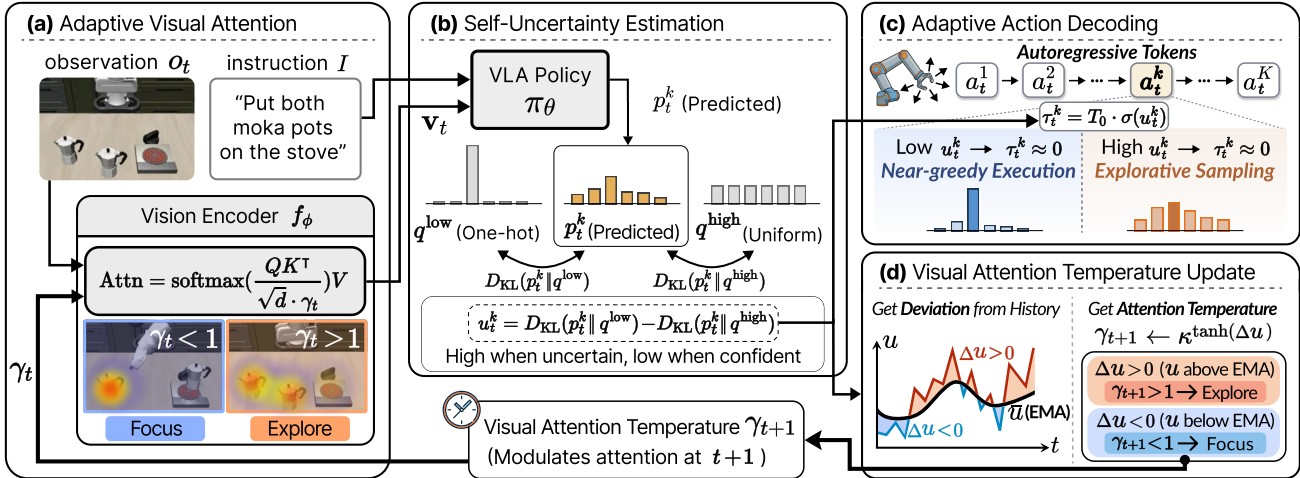

*Figure 2.* Overview of SCALE. (a) **Adaptive Visual Attention** modulates the vision encoder's attention temperature $\gamma_t$ based on uncertainty deviation from recent history—sharpening focus when confident ($\gamma_t < 1$) and broadening exploration when uncertain ($\gamma_t > 1$). (b) **Self-Uncertainty Estimation** quantifies self-uncertainty $u^k$ by measuring where the predicted distribution $p_t^k$ lies relative to two reference distributions: a one-hot $q^{\text{low}}$ (full certainty) and uniform $q^{\text{high}}$ (full ambiguity). (c) **Adaptive Action Decoding** scales sampling temperature $\tau^k$ based on token-level uncertainty $u^k$—enabling near-greedy execution under confidence and diverse sampling under ambiguity. (d) **Visual Attention Temperature Update** compares the current step-level uncertainty $u_t$ against its recent history (EMA, $\bar{u}_{t-1}$) to obtain deviation $\Delta u_t := u_t - \bar{u}_{t-1}$, then converts it into attention temperature $\gamma_{t+1}$—when $u_t$ exceeds the EMA ($\Delta u_t > 0$), $\gamma_{t+1} > 1$ broadens attention (explore); when below ($\Delta u_t < 0$), $\gamma_{t+1} < 1$ sharpens attention (focus).

2018) and top-$p$ (Holtzman et al., 2020), while recent methods adaptively adjust temperature based on entropy (Zhang et al., 2024) or token difficulty (Zhu et al., 2024; Nguyen et al., 2025; Basu et al., 2021). Beyond decoding, uncertainty has guided reasoning path selection for test-time scaling (Fu et al., 2025), with extensions to VLMs (Fang et al., 2025) and VLAs (Valle et al., 2025; Zollo & Zemel, 2025; Gu et al., 2025; Römer et al., 2025). However, these VLA methods leverage uncertainty for failure prediction or calibration analysis, rather than modulating inference behavior. Recently, Self-certainty (Kang et al., 2025) in LLMs proposed distributional confidence, measuring divergence from a uniform distribution to estimate uncertainty from output logits—offering a training-free alternative to methods requiring external modules. However, this formulation captures only overall distributional uncertainty, not how confidently the model commits to its top-1 choice—critical for VLAs where greedy decoding selects the top-1 action for immediate execution. We address this by proposing a dual-reference measure that captures both distributional spread and top-1 decisiveness, leveraging it for jointly modulating perception and action in VLAs.

**Visual attention for VLMs and VLAs.** Effectively allocating attention to task-relevant image regions is crucial for VLM accuracy and hallucination mitigation (Zhang et al., 2025a; Chen et al., 2025). This extends to VLAs, where focusing on manipulation-relevant regions improves performance (Wu et al., 2025; Zhang et al., 2025b; Xiao et al., 2025; Song et al., 2025). However, existing methods rely

on contrastive masking or trained modules to regulate visual processing. In contrast, we propose a training-free approach that dynamically modulates visual attention during execution—broadening exploration under uncertainty and sharpening focus under confidence—enabling adaptive perception specifically suited for closed-loop VLA control.

## 3. Approach

To enhance VLA robustness at test time without external verifiers or multiple rollouts, we present SCALE (**S**elf-uncertainty **C**onditioned **A**daptive **L**ooking and **E**xecution), a single-pass adaptive inference strategy that jointly modulates action decoding and visual attention based on the model's own predictive uncertainty (Fig. 2). Our key insight is that self-uncertainty derived from the output distribution of VLA models can serve as an intrinsic signal for balancing exploitation and exploration—inspired by a principle of *Active Inference theory* (Friston et al., 2016), in which agents reduce uncertainty by adapting both perception and action under ambiguity. Below, we first formalize the problem setup (Sec. 3.1), then introduce our self-uncertainty measure (Sec. 3.2), and finally describe how SCALE leverages this self-uncertainty signal for adaptive inference (Sec. 3.3).

### 3.1. Preliminaries and Motivation

We consider autoregressive VLA policies $\pi_\theta$ that, at each timestep $t$, predict actions discretized into a sequence of $K$ tokens $\mathbf{a}_t = (a_t^1, \ldots, a_t^K)$. Each action is mapped to a

discrete token from an action vocabulary $\mathcal{V}$, allowing $\pi_\theta$ to factorize as:

$$\pi_\theta(\mathbf{a}_t \mid \mathbf{v}_t, I) = \prod_{k=1}^{K} \pi_\theta\big(a_t^k \mid \mathbf{v}_t, I, a_t^{<k}\big), \quad (1)$$

where $I$ is the language instruction, $a^{<k}$ denotes previously decoded tokens, and $\mathbf{v}_t = f_\phi(o_t; \gamma)$ is the visual representation obtained by processing the raw observation $o_t$ through a Transformer-based vision encoder $f_\phi$ (*e.g.*, SigLIP (Zhai et al., 2023)) with attention temperature $\gamma$ (default $\gamma = 1$).

At each token position $k$, the $\pi_\theta$ produces the logit vector $\ell_t^k \in \mathbb{R}^{|\mathcal{V}|}$, yielding a categorical distribution $p_t^k = \mathrm{softmax}(\ell_t^k)$, and we denote the top-1 probability as $p_{t,\max}^k = \max_{x \in \mathcal{V}} p_t^k(x)$.

In practice, most autoregressive VLAs follow a fixed inference pipeline: the frozen vision encoder processes observations, and greedy decoding selects the top-1 token at each step (Kim et al., 2024; Qu et al., 2025). While effective in many cases, this fixed pipeline may struggle under ambiguous situations—*perceptual ambiguity* (*i.e.*, visually similar distractors are present) or *action multimodality* (*i.e.*, multiple plausible actions exist for a given state)—where greedy decoding overlooks viable alternatives, and fixed visual processing may miss task-relevant cues, as suggested by work on active perception in robotics (Bajcsy, 1988; Bohg et al., 2017) and attention analysis in VLMs (Chen et al., 2025). In closed-loop control, where each action influences subsequent observations, such rigidity can compound mistakes over time (Ross et al., 2011).

### 3.2. Self-Uncertainty via Distributional Positioning

To move beyond the rigidity of fixed inference, we aim to adaptively modulate both action and perception based on the model's internal uncertainty—broadening exploration under ambiguity while focusing on exploitation when confident. As motivated in Sec. 1, such modulation requires a measure that captures both overall distributional uncertainty and decisiveness in the top-1 action; we achieve this by comparing distances to opposite ends of the certainty spectrum.

Specifically, inspired by log-likelihood ratio testing (Neyman & Pearson, 1933; Kullback & Leibler, 1951), which compares two competing hypotheses by measuring their relative likelihood, we define two reference distributions representing each extreme:

- **Low-uncertainty reference** ($q^{\mathrm{low}}$): A one-hot distribution on the model's top-1 token, representing full certainty—complete commitment to its current choice (see Appendix A for discussion). Implemented as $q^{\mathrm{low}}(x) = 1 - \epsilon$ for $x = \arg\max p_t^k$ and $\frac{\epsilon}{|\mathcal{V}|-1}$ otherwise, with small $\epsilon$ for numerical stability (Appendix L).

- **High-uncertainty reference** ($q^{\mathrm{high}}$): A uniform distribution $q^{\mathrm{high}}(x) = 1/|\mathcal{V}|$ for all $x \in \mathcal{V}$, representing full ambiguity—complete distributional uncertainty.

The self-uncertainty $u^k$ at position $k$ is then defined as:

$$u_t^k = D_{\mathrm{KL}}\big(p_t^k \,\|\, q^{\mathrm{low}}\big) - D_{\mathrm{KL}}\big(p_t^k \,\|\, q^{\mathrm{high}}\big). \quad (2)$$

This formulation compares how well each reference explains the predicted distribution, effectively positioning it on the certainty spectrum between full certainty and full ambiguity.

**Interpretation.** Expanding Equation 2 yields:

$$u_t^k = \mathbb{E}_{x \sim p_t^k} \left[ \log \frac{q^{\mathrm{high}}(x)}{q^{\mathrm{low}}(x)} \right], \quad (3)$$

which is precisely the *expected log-likelihood ratio* between the two references under the model's own predictive distribution (see Appendix B for derivation). Intuitively, $u_t^k > 0$ indicates the distribution lies closer to full ambiguity than full certainty, signaling high uncertainty.

This formulation inherits the benefits of distributional approaches (Kang et al., 2025)—requiring only output logits without additional training—while directly reflecting top-1 confidence through $q^{\mathrm{low}}$. We empirically show that this measure outperforms existing uncertainty proxies (Sec. 4.2.1) and consistently improves diverse VLA backbones (Sec. 4.2).

### 3.3. SCALE: Uncertainty-Driven Adaptive Inference

Having defined our self-uncertainty $u_t^k$, we leverage it to jointly modulate action decoding and visual attention, following a simple design principle: explore broadly under uncertainty, focus sharply under confidence.

#### 3.3.1. ADAPTIVE ACTION DECODING

To realize this principle at the action level, we dynamically adjust the sampling temperature $\tau_t^k$ of VLA policy $\pi_\theta$ based on action token-level self-uncertainty:

$$\tau_t^k = T_0 \cdot \sigma(u_t^k), \quad (4)$$

where $T_0$ is the maximum temperature defining the exploration range, and the sigmoid $\sigma(u_t^k)$ acts as a gate that adjusts exploration based on situational uncertainty—high uncertainty opens the gate for diverse sampling, low uncertainty closes it for focused execution.

Since $u_t^k$ can be interpreted as a log-likelihood ratio between the hypotheses "uncertain" and "confident", applying the sigmoid function recovers the posterior probability of the uncertain hypothesis: $\sigma(u_t^k) = P(\mathrm{uncertain} \mid p_t^k)$ (see Appendix C for details), serving as a soft gate (Hochreiter & Schmidhuber, 1997; Dauphin et al., 2017) that yields

$\tau \approx 0$ (near-greedy) under low uncertainty and $\tau \approx T_0$ (explorative) under high uncertainty.

The action token is then sampled from the temperature-scaled distribution:

$$a_t^k \sim \text{Cat}(\text{softmax}(\ell_t^k / \tau_t^k)), \qquad (5)$$

where $\text{Cat}$ denotes the categorical distribution.

### 3.3.2. ADAPTIVE VISUAL ATTENTION

At the perception level, the same principle applies: broaden attention under uncertainty to gather information, sharpen it under confidence for focused execution (Bajcsy, 1988; Bohg et al., 2017). This modulation can occur at two points: the vision encoder $f_\phi$, which controls *what* visual information is extracted via uni-modal attention (Zou et al., 2024), or the VLA backbone $\pi_\theta$, which controls *which* extracted features are attended to via cross-modal attention (Chen et al., 2025). We choose to modulate the vision encoder, as it is the first stage that directly determines what visual information to extract—cross-modal attention can only select from what has already been encoded; we empirically compare both strategies in Sec. 4.2.1 and find that vision encoder modulation is more effective for VLAs.

Given this choice, the remaining question is *how* uncertainty should guide this modulation. For action decoding, each token's instantaneous uncertainty $u^k$ directly determines its sampling temperature without considering temporal context. Visual modulation, however, must respond to evolving scene conditions across timesteps (Bajcsy, 1988), as perception inherently relies on temporal context to interpret the current observation (Rao & Ballard, 1999; Clark, 2013). We therefore argue that comparing current uncertainty to recent history—rather than using its instantaneous value alone—better captures transitions in scene complexity; we validate this empirically in Sec. 4.2.1.

To implement this, we first aggregate token-level uncertainties into a step-level uncertainty $u_t$ by averaging following prior work (Kang et al., 2025), as visual modulation operates at the step-level:

$$u_t = \frac{1}{K} \sum_{k=1}^K u_t^k. \qquad (6)$$

We then maintain an exponential moving average (EMA) of recent uncertainty:

$$\bar{u}_t = \alpha \bar{u}_{t-1} + (1 - \alpha) u_t, \qquad (7)$$

where hyperparameter $\alpha$ controls temporal smoothing.

We detect transitions in scene complexity via the deviation from this average. In particular, for efficient single-pass execution, we modulate visual attention at timestep $t$ using

---

**Algorithm 1** SCALE; Adaptive Looking and Execution

1: **Input:** Observation $o_t$, instruction $I$, uncertainty deviation $\Delta u_{t-1}$
2: **Output:** Sequence of action tokens $\mathbf{a}_t$, uncertainty deviation $\Delta u_t$
3:
4: *// Adaptive visual attention (Sec. 3.3.2)*
5: $\gamma_t \leftarrow \kappa^{\tanh(\Delta u_{t-1})}$
6: $\mathbf{v}_t \leftarrow f_\phi(o_t; \gamma_t)$      *// Visual attention scaled by $\gamma_t$*
7:
8: *// Adaptive action decoding (Sec. 3.3.1)*
9: **for** $k = 1$ **to** $K$ **do**
10:     $\ell_t^k \leftarrow \pi_\theta(\mathbf{v}_t, I, a_t^{<k})$
11:     $p_t^k \leftarrow \text{softmax}(\ell_t^k)$
12:     $u_t^k \leftarrow D_{\text{KL}}(p_t^k \| q^{\text{low}}) - D_{\text{KL}}(p_t^k \| q^{\text{high}})$   *// Token-level uncertainty (Sec. 3.2)*
13:     $\tau_t^k \leftarrow T_0 \cdot \sigma(u_t^k)$
14:     $a_t^k \sim \text{Cat}(\text{softmax}(\ell_t^k / \tau_t^k))$
15: **end for**
16:
17: $u_t \leftarrow \frac{1}{K} \sum_{k=1}^K u_t^k$     *// Step-level uncertainty*
18: $\bar{u}_t \leftarrow \alpha \bar{u}_{t-1} + (1 - \alpha) u_t$     *// Update EMA*
19: $\Delta u_t \leftarrow u_t - \bar{u}_{t-1}$   *// Compute uncertainty deviation*
20: **Return:** $(a^1, \ldots, a^K), \Delta u_t$

---

the preceding step's deviation $\Delta u_{t-1} = u_{t-1} - \bar{u}_{t-2}$, since $u_t$ is only available after action decoding, which occurs downstream of visual encoding. We posit that consecutive visual frames are highly correlated (Bovik, 2010), making uncertainty at $t - 1$ a reliable proxy for timestep $t$; empirically, using $u_t$ with an additional forward pass yields similar performance (see Appendix D), supporting this assumption.

We convert this deviation into an attention temperature $\gamma_t$ for modulating the vision encoder's attention. Following prior work (Dinh et al., 2017), we obtain $\gamma_t$ by applying $\tanh$ and exponentiation to the deviation, yielding a value centered at 1 so that zero deviation produces no modulation:

$$\gamma_t = \kappa^{\tanh(\Delta u_{t-1})}, \qquad (8)$$

where $\kappa > 1$ bounds $\gamma_t \in (1/\kappa, \kappa)$ for stability.

We then apply $\gamma_t$ across all layers of the vision encoder $f_\phi$ to scale the self-attention, following Zou et al. (2024):

$$\text{Attn}(Q, K, V) = \text{softmax}\left(\frac{QK^\top}{\sqrt{d} \cdot \gamma_t}\right) V. \qquad (9)$$

This yields $\gamma_t > 1$ (flattened attention for broader exploration) when uncertainty rises above its recent average, and $\gamma_t < 1$ (sharpened attention for focused perception) when it falls below, as shown in Fig. 3.

Together with adaptive action decoding (Sec. 3.3.1), the entire procedure requires only a *single forward pass* per

*Table 1.* SR (%) on LIBERO with OpenVLA backbone. All methods, including ours, use OpenVLA fine-tuned on LIBERO as the base model. Results for TTS methods are taken from their respective papers; all others are reproduced in our experiments. *denotes results reproduced using authors' implementation with greedy decoding. '–' indicates results not reported in prior work. See Appendix K for standard deviations.

| Method | Spatial | Object | Goal | Long | Avg. |
|---|---|---|---|---|---|
| *Training-required, test-time scaling* | | | | | |
| RoboMonkey | – | – | – | 56.5 | – |
| TACO | – | – | – | 60.0 | – |
| MG-Select | 81.7 | 72.5 | 73.6 | 55.4 | 70.8 |
| *Training-free, single inference* | | | | | |
| OpenVLA* (fine-tuned) | 86.2 | 86.2 | 77.7 | 52.7 | 75.7 |
| + Sampling ($t$=1.0) | 85.1 | 87.9 | 78.9 | 54.7 | 76.7 |
| + Top-$k$ ($k$=40, $t$=0.7) | 85.2 | 88.2 | 78.3 | 55.2 | 76.7 |
| + Top-$p$ ($p$=0.9) | 86.9 | 88.1 | 78.6 | 55.1 | 77.2 |
| + SCALE (Ours) | **89.5** | **91.0** | **82.3** | **63.3** | **81.5** |

*Table 2.* SR (%) on LIBERO with $\pi_0$-FAST backbone fine-tuned on LIBERO. *reproduced with greedy decoding. See Appendix K for standard deviations.

| Method | Spatial | Object | Goal | Long | Avg. |
|---|---|---|---|---|---|
| $\pi_0$-FAST * (fine-tuned) | 96.6 | 98.1 | 93.7 | 76.3 | 91.2 |
| + Sampling ($t$=1.0) | 87.0 | 94.6 | 83.5 | 72.2 | 84.3 |
| + Top-$k$ ($k$=40, t=0.7) | 93.7 | 96.5 | 87.5 | 74.8 | 88.1 |
| + Top-$p$ ($p$=0.9) | 90.2 | 95.3 | 85.9 | 73.4 | 86.2 |
| + SCALE (Ours) | **97.7** | **98.7** | **94.7** | **80.9** | **93.0** |

*Table 3.* SR (%) on SIMPLER-WidowX with $\pi_0$-FAST and SpatialVLA backbones. $\pi_0$-FAST and SpatialVLA (fine-tuned) are trained on BridgeData V2; SpatialVLA (zero-shot) is trained on OXE (O'Neill et al., 2024) and evaluated zero-shot. *reproduced with official implementation using greedy decoding.

| Method | Spoon | Carrot | Cube | Eggplant | Avg. |
|---|---|---|---|---|---|
| $\pi_0$-FAST * (fine-tuned) | 20.8 | 62.5 | 37.5 | 16.7 | 34.4 |
| + Sampling ($t$=1.0) | 41.7 | 54.2 | 29.2 | 16.7 | 35.4 |
| + Top-$k$ ($k$=40, t=0.7) | 48.6 | 61.1 | 41.7 | 15.3 | 41.7 |
| + Top-$p$ ($p$=0.9) | 48.6 | 62.5 | 33.3 | 16.7 | 40.3 |
| + SCALE (Ours) | **58.3** | **69.4** | **48.6** | **19.4** | **49.0** |
| SpatialVLA* (fine-tuned) | 20.8 | 25.0 | 20.8 | **100.0** | 41.7 |
| + Sampling ($t$=1.0) | 18.1 | 20.8 | 25.0 | **100.0** | 41.0 |
| + Top-$k$ ($k$=40, t=0.7) | 12.5 | 23.6 | 20.8 | **100.0** | 39.2 |
| + Top-$p$ ($p$=0.9) | 18.1 | 23.6 | 25.0 | 98.6 | 41.3 |
| + SCALE (Ours) | **22.2** | **31.9** | **26.4** | **100.0** | **45.1** |
| SpatialVLA* (zero-shot) | 12.5 | 20.8 | 25.0 | 66.7 | 31.3 |
| + Sampling ($t$=1.0) | 16.7 | 23.6 | 22.2 | 66.7 | 32.3 |
| + Top-$k$ ($k$=40, t=0.7) | 8.3 | 29.2 | 20.8 | 61.1 | 29.9 |
| + Top-$p$ ($p$=0.9) | 13.9 | 26.4 | 22.2 | 61.1 | 30.9 |
| + SCALE (Ours) | **22.2** | **34.7** | **34.7** | **75.0** | **41.7** |

*Table 4.* SR (%) on LIBERO-PRO-Long under various perturbations with OpenVLA and $\pi_0$-FAST backbones. Both models are trained on LIBERO and evaluated zero-shot. *reproduced with official implementation using greedy decoding.

| Method | Language | Object | Task | Swap | Avg. |
|---|---|---|---|---|---|
| OpenVLA* (zero-shot) | 42.0 | 26.6 | 3.2 | 0.0 | 18.0 |
| + Sampling ($t$=1.0) | 44.8 | 27.6 | 3.2 | 0.0 | 18.9 |
| + Top-$k$ ($k$=40, t=0.7) | 44.6 | 26.0 | 4.0 | 0.0 | 18.7 |
| + Top-$p$ ($p$=0.9) | 44.2 | 28.2 | 3.6 | 0.0 | 19.0 |
| + SCALE (Ours) | **51.2** | **30.0** | **4.8** | 0.0 | **21.5** |
| $\pi_0$-FAST * (zero-shot) | 78.0 | 49.0 | 13.6 | 2.2 | 35.7 |
| + Sampling ($t$=1.0) | 74.8 | 46.2 | 13.8 | 2.4 | 34.3 |
| + Top-$k$ ($k$=40, t=0.7) | 75.6 | 44.6 | 14.6 | 2.6 | 34.4 |
| + Top-$p$ ($p$=0.9) | 75.4 | 44.8 | 14.2 | 2.6 | 34.3 |
| + SCALE (Ours) | **84.0** | **51.8** | **15.8** | **3.4** | **38.8** |

control step: uncertainty is computed from logits during action decoding, and visual modulation reuses the previous step's uncertainty—requiring no additional rollouts, external verifiers, or auxiliary training (see Appendix E for cost comparison). Algorithm 1 summarizes the procedure.

## 4. Experiments

### 4.1. Experimental Setups

We evaluate SCALE in both simulation and real-world settings using multiple autoregressive VLA backbones. In simulation, we use OpenVLA (Kim et al., 2024), $\pi_0$-FAST (Pertsch et al., 2025), and SpatialVLA (Qu et al., 2025); in real-world experiments, we use OpenVLA and $\pi_0$-FAST. We build upon the authors' official codebases[1] to reproduce baselines, applying SCALE and other decoding strategies on top of the same implementations. Each model is evaluated either fine-tuned or zero-shot depending on the benchmark, as indicated in each table. For each backbone, we select the hyperparameters of SCALE using

only LIBERO-Long and apply the same configuration to all other simulation benchmarks and real-world experiments without further adjustment, ensuring a fair, generalizable evaluation. See Appendix G for full implementation details and Appendix L for hyperparameter sensitivity analysis.

**Simulation benchmarks.** We use three complementary benchmarks: **LIBERO** (Liu et al., 2023) for multi-task generalization across object, layout, goal, and long-horizon variations; **SIMPLER-WidowX** (Li et al., 2024) for execution precision in real-to-sim pick-and-place; and **LIBERO-PRO-Long** (Zhou et al., 2025), the most challenging split, as an *unseen* benchmark for robustness beyond memorization (see Appendix F.1 for more details about benchmarks).

**Real-world setup.** Following prior work (Jang et al., 2025), we evaluate under both *in-distribution* (ID) and *out-*

---

[1]Official repositories: openvla/openvla, Physical-Intelligence/openpi, and SpatialVLA/SpatialVLA, all on GitHub.

*Table 5.* SR (%) on real-world "Put A on B" pick-and-place tasks (A/B = object/receptacle) under ID and OOD conditions. We compare SCALE against greedy decoding on OpenVLA and $\pi_0$-FAST backbones fine-tuned on our teleoperated demonstrations.

| Method | In-Distribution | | | | Out-of-Distribution | | |
|---|---|---|---|---|---|---|---|
| | Carrot/Towel | Eggplant/Bowl | Lemon/Plate | Avg. | Teddy Bear/Bowl | Cube/Plate | Avg. |
| OpenVLA | 45.8 | 45.8 | 16.7 | 36.1 | 29.2 | 16.7 | 22.9 |
| + SCALE (Ours) | **75.0** | **62.5** | **29.2** | **55.6** | **45.8** | **33.3** | **39.6** |
| $\pi_0$-FAST | 66.7 | 75.0 | 75.0 | 72.2 | 37.5 | 50.0 | 43.8 |
| + SCALE (Ours) | **87.5** | **87.5** | **83.3** | **86.1** | **50.0** | **62.5** | **56.3** |

*Table 6.* We evaluate the contribution of adaptive decoding (Ada. Decoding) and adaptive visual attention (Ada. Visual Attention). Both components provide complementary gains, achieving the best performance when combined. Without adaptive decoding, we use greedy decoding by default. The last row corresponds to SCALE.

| Ada. Decoding | Ada. Visual Attention. | SR (%) |
|---|---|---|
| ✗ | ✗ | 52.7 |
| ✓ | ✗ | 58.0 |
| ✗ | ✓ | 56.0 |
| ✓ | ✓ | **63.3** |

*Table 7.* Comparison of different uncertainty metrics for adaptive decoding and perception. Confidence-based methods are inverted to represent *uncertainty*. All variants use the same adaptive action decoding and visual attention modulation pipeline.

| Metric | SR (%) |
|---|---|
| Baseline; OpenVLA | 52.7 |
| *Confidence-based* | |
| $p_{max}$ (Hendrycks & Gimpel, 2017) | 56.0 |
| Self-certainty (Kang et al., 2025) | 53.8 |
| *Uncertainty-based* | |
| Gini Impurity (Breiman et al., 1984) | 57.8 |
| Entropy (Malinin & Gales, 2021) | 55.4 |
| **Self-uncertainty (Ours)** | **63.3** |

*Table 8.* Design choices for visual modulation. We compare three dimensions: (1) modulation target—uni-modal attention (attn.) in vision encoder $f_\phi$ *vs.* cross-modal attention in VLA $\pi_\theta$; (2) modulation strategy—Fixed (w/ sign; binary switch at $u_{t-1} = 0$) *vs.* Adaptive (continuous scaling); and (3) uncertainty signal—instantaneous ($u_{t-1}$) *vs.* change-based ($\Delta u_{t-1}$). All variants include adaptive action decoding. The last row shows SCALE.

| Modulation Target | Strategy | Signal | SR (%) |
|---|---|---|---|
| Baseline; OpenVLA | — | — | 52.7 |
| $\pi_\theta$ cross-modal attn. | Fixed | $sign(u_{t-1})$ | 54.8 |
| $\pi_\theta$ cross-modal attn. | Adaptive | $\Delta u_{t-1}$ | 57.4 |
| $f_\phi$ uni-modal attn. | Adaptive | $u_{t-1}$ | 55.4 |
| $f_\phi$ uni-modal attn. | Adaptive | $\Delta u_{t-1}$ | **63.3** |

*decoding*: temperature (Radford et al., 2019), top-$k$ (Fan et al., 2018), and top-$p$ sampling (Holtzman et al., 2020). Hyperparameters are detailed in the tables, with sensitivity analysis in Appendix H. For real-world experiments, we compare against greedy decoding, as training-free alternatives show similar performance in simulation environment. We report *success rate* (SR, %) as the metric, averaged over three seeds for simulation and 24 episodes per task for real-world evaluation.

### 4.2. Quantitative Analyses

Tables 1–5 summarize our main results across simulation and real-world benchmarks. We highlight three key findings.

**(1) SCALE consistently improves over greedy decoding across all benchmarks and backbones.** In simulation, SCALE achieves gains on LIBERO (+5.8 avg. with Open-VLA, +1.8 with $\pi_0$-FAST; Tables 1, 2), SIMPLER-WidowX (+3.4/+10.4 with SpatialVLA fine-tuned/zero-shot, +14.6 with $\pi_0$-FAST; Table 3), and LIBERO-PRO-Long (+3.5 with OpenVLA, +3.1 with $\pi_0$-FAST; Table 4). In real-world experiments (Table 5), gains are more pronounced under both in-distribution (+19.5 with OpenVLA, +13.9 with $\pi_0$-FAST) and out-of-distribution conditions (+16.7 with OpenVLA, +12.5 with $\pi_0$-FAST). These results demonstrate that SCALE generalizes across architectures, task com-

*of-distribution* (OOD) conditions using a 6-DoF UR10e arm equipped with a Robotiq 2F-85 gripper. ID tasks consist of three "Put A on B" pick-and-place tasks involving objects of different geometries (*e.g.*, carrot, eggplant, lemon); OOD tasks follow a similar task setup, but introduce unseen objects with more challenging geometries and compliances (*e.g.*, soft teddy bear, small cube). We fine-tune OpenVLA and $\pi_0$-FAST on 48 teleoperated demonstrations per ID task prior to evaluation (see Appendix F.2).

**Baselines and evaluation metric.** For simulation, we compare against: (1) *Training-required TTS*: RoboMonkey (Kwok et al., 2025), TACO (Yang et al., 2025), and MG-Select (Jang et al., 2025), which perform Best-of-$N$ selection with external or self-verification; (2) *Training-free*

**Task:** Put the yellow and white mug in the microwave

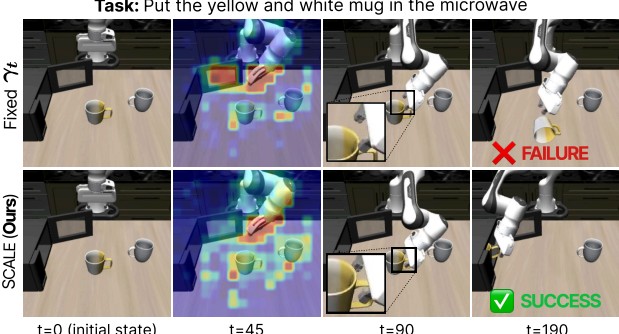

t=0 (initial state)  t=45  t=90  t=190

**Task:** Put the eggplant in the bowl

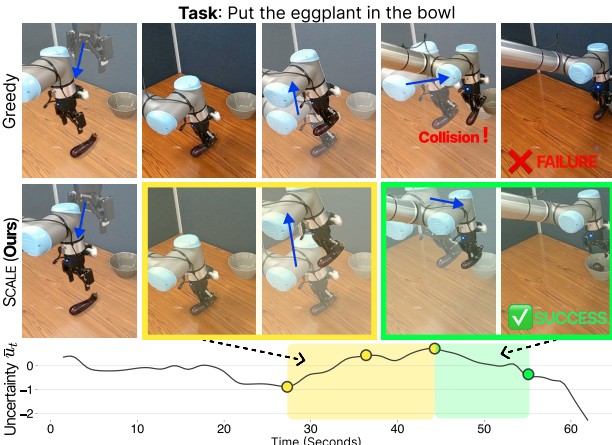

*Figure 3.* Qualitative result of adaptive visual attention. We visualize attention maps from SigLIP, the vision encoder $f_\phi$ of Open-VLA, at $t=45$, where self-uncertainty increases. Color indicates attention intensity (blue: low, green: medium, red: high). The fixed-temperature baseline ($\gamma_t=1$) attends to task-irrelevant regions such as the microwave door, whereas SCALE increases $\gamma_t$ to broaden attention and allocate more focus to the target mug. This uncertainty-driven modulation improves visual grounding and leads to successful grasping in subsequent steps.

*Figure 4.* Qualitative result of adaptive action decoding. We compare greedy decoding (top) and SCALE (middle) on a real-world manipulation task using $\pi_0$-FAST; blue arrows indicate robot motion. Greedy decoding follows a direct path and collides with the bowl, whereas SCALE uses high self-uncertainty to broaden action exploration and select an elevated trajectory that clears the obstacle. The bottom plot shows the temporal dynamics of the smoothed self-uncertainty $\bar{u}_t$, which increases during the critical transport phase and decreases once the robot reaches a stable trajectory toward task completion.

plexities, and deployment settings.

**(2) Naive decoding strategies often *degrade* performance, while SCALE provides robust improvements.** On LIBERO with $\pi_0$-FAST (Table 2), temperature sampling, top-$k$, and top-$p$ sampling all underperform greedy decoding ($91.2 \rightarrow 84.3/88.1/86.2$). We attribute this to fixed hyperparameters that cannot adapt to varying uncertainty across states and tasks; indeed, no single setting consistently outperforms greedy decoding across all benchmarks (Appendix H). In contrast, SCALE dynamically adjusts sampling temperature based on the model's predicted uncertainty, achieving robust improvements.

**(3) SCALE outperforms training-required TTS methods while remaining training-free and single-pass.** On LIBERO with OpenVLA (Table 1), SCALE outperforms MG-Select by +10.7 points on average (81.5 *vs.* 70.8), with a notable gap on long-horizon tasks (63.3 *vs.* 55.4). SCALE also surpasses RoboMonkey and TACO on LIBERO-Long by +6.8 and +3.3 points, respectively (see Appendix I for per-task breakdown). This demonstrates that adaptive modulation achieves superior performance without external verification or additional forward passes.

### 4.2.1. DETAILED ANALYSES

For detailed analyses, we use OpenVLA (Kim et al., 2024) on the LIBERO-Long benchmark (Liu et al., 2023), as it provides challenging long-horizon tasks well suited to evaluate adaptive inference across diverse scenarios (Appendix I).

**Ablation study.** Table 6 shows the contribution of each component in SCALE. Each component alone improves

over the baseline (rows 2–3 *vs.* row 1): adaptive decoding yields +5.3 ($52.7 \rightarrow 58.0$) and adaptive visual attention +3.3 ($52.7 \rightarrow 56.0$). Yet combining both (row 4; SCALE) reaches 63.3 ($+10.6$), exceeding the sum of the individual gains ($+8.6$) by over $2\%$. This shows the two are not merely complementary but *synergistic*: a coupled loop in which improved perception yields better uncertainty estimates that guide action decoding, and vice versa—operationalizing the joint perception–action adaptation of Active Inference theory (Friston et al., 2016) for VLAs.

**Self-uncertainty measure.** We compare our self-uncertainty against alternatives (Tab. 7): $p_{\max}$, Gini impurity, Entropy and Self-Certainty. For fair comparison, we normalize each metric to $[0,1]$—inverting confidence-based scores (*e.g.*, $p_{\max}$ and Self-Certainty) to represent uncertainty—and apply the same inference pipeline: replacing $\sigma(u_t^k)$ for action decoding and using the momentum-based scheme (Eq. 8) for visual attention (see Appendix J for details). While all uncertainty measures improve over the baseline (row 1), our formulation achieves the highest success rate, outperforming the next best by 5.5. Notably, Self-Certainty (Kang et al., 2025), which captures only distributional uncertainty, yields marginal gains. These results suggest that our dual-reference measure is better suited for VLAs, as it captures both uncertainty and decisiveness, simultaneously—essential for adaptive looking and execution.

**Design choices for visual modulation.** Tab. 8 compares three design axes: (1) the modulation target—uni-modal attention in the vision encoder $f_\phi$ *vs.* cross-modal attention in the VLA backbone $\pi_\theta$; (2) the modulation strategy—fixed (binary switching on $\text{sign}(u_{t-1})$) *vs.* adaptive (continuous scaling); and (3) the uncertainty signal—instantaneous ($u_{t-1}$) *vs.* change-based ($\Delta u_{t-1}$). Every configuration improves over the greedy baseline (52.7), and combining the best option along each axis (last row) yields SCALE (63.3). Modulating uni-modal attention outperforms cross-modal attention (63.3 *vs.* 57.4, +5.9), confirming that adjusting *what* is captured before fusion is more effective than reweighting *how* already-encoded features are integrated. Adaptive scaling outperforms fixed switching (57.4 *vs.* 54.8, +2.6), as matching modulation strength to the uncertainty level provides finer control than a binary switch. Finally, the change-based signal $\Delta u_{t-1}$ surpasses the instantaneous $u_{t-1}$ by the widest margin (63.3 *vs.* 55.4, +7.9), supporting our hypothesis that perception should respond to *transitions* in scene complexity rather than absolute uncertainty.

### 4.3. Qualitative Analysis

Figures 3 and 4 illustrate the effect of adaptive visual attention and action decoding, respectively. For visual attention (Fig. 3), we visualize attention from SigLIP, the vision encoder of OpenVLA. At $t=45$ when self-uncertainty suddenly increases, the baseline with fixed $\gamma_t=1$ attends to task-irrelevant regions such as the microwave door while underattending to the target mug, leading to task failure. In contrast, SCALE dynamically increases $\gamma_t$ to broaden attention across the scene, redirecting focus from the microwave door to the task-relevant mug—enabling a successful grasp ($t=90$) and eventual task completion. For action decoding (Fig. 4), the bottom plot shows $\bar{u}_t$ temporal dynamics: initially high due to multiple viable options, dropping during grasping, then rising while transporting the object. Greedy decoding follows a direct path and collides with the bowl, whereas SCALE leverages high uncertainty to find an elevated trajectory that clears the bowl (yellow phase); once the robot reaches a stable position, uncertainty drops, leading to task success (green phase). These results demonstrate that adaptive modulation of both perception and action is effective for robust closed-loop control.

## 5. Conclusion

We presented SCALE, a simple inference strategy that enhances VLA robustness by jointly modulating perception and action based on self-uncertainty—requiring no additional training, no external verifier, and only a single forward pass. Inspired by *uncertainty-driven exploration* in Active Inference theory, SCALE broadens exploration under ambiguity while focusing on exploitation when confident.

Central to our approach is a self-uncertainty measure that compares distances to both extremes of the certainty spectrum, capturing both distributional concentration and top-1 confidence. Experiments on simulated and real-world benchmarks show that SCALE consistently improves SoTA VLAs and outperforms existing TTS methods.

## Acknowledgements

This work was partly supported by the InnoCORE program (26-InnoCORE-01), the IITP grants (RS-2022-II220077, RS-2022-II220113, RS-2022-II220959, RS-2022-II220871, RS-2026-25507282, RS-2026-25518317, RS-2021-II211343 (SNU AI), RS-2025-25442338 (AI Star Fellowship-SNU)) and Advanced GPU utilization support program (02-26-01-0285) (via NIPA) funded by the Korea government (MSIT), grants (RS-2025-25462891 (US-KOR BARI), RS-2025-25453780) funded by MOTIR, a grant of Korean ARPA-H Project through the Korea Health Industry Development Institute (KHIDI), funded by the Ministry of Health & Welfare, Republic of Korea (RS-2025-25424639), and the BK21 FOUR program, SNU in 2025.

## Impact Statement

This paper presents SCALE, a test-time inference strategy for Vision-Language-Action models in robotic control. Our work aims to enhance the robustness of embodied AI systems by enabling adaptive perception and action under uncertainty, potentially improving the safety and reliability of robots operating in diverse real-world environments.

The broader implications of more capable robotic systems include both benefits—such as improved automation in manufacturing, healthcare, and assistive technologies—and potential concerns around workforce displacement and safety in human-robot interaction. However, as our contribution is primarily methodological and focuses on improving existing VLA architectures without introducing new capabilities, we do not anticipate unique ethical concerns beyond those inherent to the broader field of robot learning. We encourage practitioners deploying such systems to carefully consider safety protocols and human oversight in their applications.

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

# A. On the Design of Low-Uncertainty Reference

A potential concern with our low-uncertainty reference $q^{\text{low}}$ is its self-referential nature: it anchors to the model's own top-1 prediction rather than an external ground truth. However, this design choice aligns with our goal.

**Goal: Measuring conviction, not correctness.** Our objective is not to assess whether the model's prediction is *correct*, but rather how *decisive* it is about its current choice. By anchoring $q^{\text{low}}$ to the model's own top-1 token, our measure directly quantifies this internal conviction—high conviction signals low uncertainty requiring less exploration, while a diffuse distribution signals ambiguity warranting broader exploration.

**Empirical validation.** For this self-referential design to be meaningful, conviction should carry task-relevant information. We analyzed the relationship between episode-level average $p_{\text{max}}$ and task success rate across 6,000 episodes on LIBERO benchmarks. As shown in Fig. 5, success rates decline sharply in the lowest $p_{\text{max}}$ regime, indicating that conviction reliably identifies high-risk trajectories and thus serves as a valid basis for adaptive control.

**Connection to Active Inference.** This self-referential design aligns with Active Inference theory (Friston et al., 2016; Schwartenbeck et al., 2019), where agents estimate uncertainty from their own generative models rather than external oracles, and reduce it by adapting both perception and action—a principle observed in humans (Daw et al., 2006; Wilson et al., 2014) and formalized in robotics as active perception (Bohg et al., 2017; Bajcsy et al., 2018). This provides theoretical grounding for why self-referential uncertainty can effectively guide adaptive behavior.

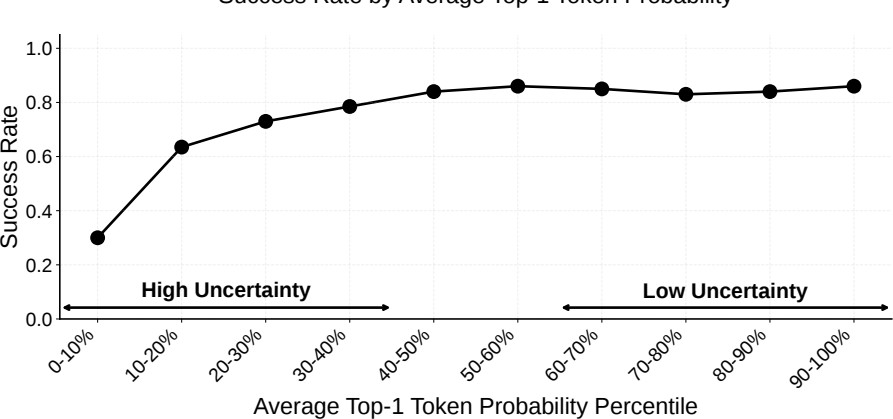

*Figure 5.* **Task success rate by average** $p_{\text{max}}$**.** Results aggregated over 6,000 episodes across LIBERO benchmarks using OpenVLA. Episodes with low average $p_{\text{max}}$ exhibit significantly lower success rates, indicating that $p_{\text{max}}$ serves as a reliable signal for the model's conviction and potential failure risk.

# B. Derivation of Self-uncertainty Formulation

In this section, we provide the derivation for the self-uncertainty metric $u_t^k$ presented in Equation 3. Recall the definition of $u_t^k$ as the difference between the KL divergence of the prediction $p_t^k$ from the low-uncertainty reference $q^{\text{low}}$ and the high-uncertainty reference $q^{\text{high}}$:

$$u_t^k = D_{\text{KL}}(p_t^k \| q^{\text{low}}) - D_{\text{KL}}(p_t^k \| q^{\text{high}}). \tag{10}$$

By expanding the KL divergence terms using the definition $D_{\text{KL}}(P\|Q) = \mathbb{E}_{x\sim P}[\log P(x) - \log Q(x)]$, we obtain:

$$u_t^k = \mathbb{E}_{x\sim p_t^k}\left[\log p_t^k(x) - \log q^{\text{low}}(x)\right] - \mathbb{E}_{x\sim p_t^k}\left[\log p_t^k(x) - \log q^{\text{high}}(x)\right] \tag{11}$$

$$= \mathbb{E}_{x\sim p_t^k}\left[(\log p_t^k(x) - \log q^{\text{low}}(x)) - (\log p_t^k(x) - \log q^{\text{high}}(x))\right] \tag{12}$$

$$= \mathbb{E}_{x\sim p_t^k}\left[\log q^{\text{high}}(x) - \log q^{\text{low}}(x)\right] \tag{13}$$

$$= \mathbb{E}_{x\sim p_t^k}\left[\log \frac{q^{\text{high}}(x)}{q^{\text{low}}(x)}\right]. \tag{14}$$

This result confirms that $u_t^k$ represents the expected log-likelihood ratio between the high-uncertainty and low-uncertainty references under the model's current predictive distribution.

## C. Probabilistic Interpretation: $\sigma(u_t^k)$ as Posterior Probability

We interpret our scaling mechanism within a binary hypothesis testing framework. Consider two hypotheses regarding the model's state: an uncertain state $H_{\text{high}}$ (modeled by $q^{\text{high}}$) and a confident state $H_{\text{low}}$ (modeled by $q^{\text{low}}$).

Using Bayes' rule, the log-odds of the posterior probability $P(H_{\text{high}} \mid p_t^k)$ can be expressed as the sum of the log-likelihood ratio (LLR) and the prior log-odds:

$$\log \frac{P(H_{\text{high}} \mid p_t^k)}{P(H_{\text{low}} \mid p_t^k)} = \underbrace{\log \frac{P(p_t^k \mid H_{\text{high}})}{P(p_t^k \mid H_{\text{low}})}}_{\text{LLR} \approx u_t^k} + \log \frac{P(H_{\text{high}})}{P(H_{\text{low}})}. \tag{15}$$

Our metric $u^k$ corresponds to the expected LLR (Kullback & Leibler, 1951). Assuming uninformative priors ($P(H_{\text{high}}) = P(H_{\text{low}})$), the prior term vanishes, and the relation simplifies to:

$$\text{logit}\big(P(H_{\text{high}} \mid p_t^k)\big) \approx u_t^k. \tag{16}$$

Since the sigmoid function is the inverse of the logit, applying it to $u_t^k$ yields the posterior probability:

$$\sigma(u_t^k) \approx P(H_{\text{high}} \mid p_t^k). \tag{17}$$

This suggests that $\sigma(u^k)$ serves as an estimate of the probability that the current state is uncertain, providing a probabilistic basis for its use in temperature scaling.

## D. Empirical Validation of Previous-Step Deviation for Single-Pass Inference

In our proposed method, SCALE, we modulate the visual attention mechanism at timestep $t$ using the uncertainty deviation derived from the *previous* timestep (i.e., $\Delta u_{t-1} = u_{t-1} - \bar{u}_{t-2}$). This design choice enables efficient, single-pass inference. However, an ideal "Oracle" modulation strategy would condition the vision encoder on the uncertainty of the *current* observation $u_t$.

To assess the trade-off between computational efficiency and modulation accuracy, following Chen et al. (2025) which addresses spatial reasoning in VLMs via confidence-based two-step inference, we implement a Two-step Inference Oracle. This baseline requires two forward passes per control step to utilize the exact current uncertainty:

1. **Probe Pass:** The model processes the observation with standard visual attention ($\gamma_t = 1$) and performs greedy decoding to compute the exact step-level uncertainty deviation $\Delta u_t = u_t - \bar{u}_{t-1}$.

2. **Execution Pass:** The vision encoder is re-run with the attention temperature $\gamma_t$ adjusted based on the computed current deviation $\Delta u_t$ from the probe pass, followed by our adaptive action decoding.

Table 9 presents the performance comparison and inference cost on the LIBERO-Long benchmark. As hypothesized, the Two-step Oracle yields the highest performance (64.6%), serving as the empirical upper bound for our modulation strategy. However, similar to test-time scaling methods, this accuracy comes at the cost of doubling the evaluation time ($\sim$26 hours vs. $\sim$13 hours), rendering it impractical for real-time robotic control applications. Crucially, SCALE achieves a success rate of 63.3%, performing within a negligible margin ($-1.3\%$) of the Oracle while maintaining the same single-pass efficiency as the base OpenVLA model. This result empirically validates our reliance on the previous step's deviation $\Delta u_{t-1}$. In high-frequency control loops, visual states and their corresponding uncertainties exhibit high temporal correlation; consequently, the uncertainty signal from the preceding step serves as a sufficiently accurate proxy for the current state, enabling SCALE to capture the benefits of adaptive looking without the computational penalty of iterative inference.

## E. Comparative Analysis of Inference and Training Efficiency

In this section, we analyze the efficiency of SCALE relative to existing test-time scaling (TTS) methods across two dimensions: inference latency and training overhead.

*Table 9.* Performance Comparison with Two-step Inference Oracle. Total evaluation time denotes the wall-clock time required for evaluating all 500 episodes of LIBERO-Long on a single NVIDIA A6000 GPU.

| Method | LIBERO-Long | Total Evaluation Time (h) |
|---|---|---|
| OpenVLA* (fine-tuned) | 52.7 | ~13 |
| + Two-step Oracle | 64.6 | ~26 |
| + SCALE **(Ours)** | 63.3 | ~13 |

**Inference Latency.** To evaluate the computational cost of generating multiple action candidates, we measured the wall-clock time for OpenVLA and $\pi_0$-FAST on the LIBERO-Spatial. We conducted evaluations across 50 episodes, recording the time required to generate action tokens as the number of samples $N$ increased. As shown in Fig. 6, the latency for both models increases significantly with $N$. Specifically, at $N = 16$, OpenVLA and $\pi_0$-FAST exhibit approximately $15.9\times$ and $3.2\times$ increases in latency compared to single-sample generation, respectively. This disparity highlights the bottleneck inherent in TTS methods that require multiple forward passes, particularly for models like OpenVLA that lack efficient batch inference support. Note that these measurements reflect only action generation latency and exclude the additional computational overhead associated with the verification process.

**Training Overhead.** Beyond inference-time rollouts, many TTS methods rely on auxiliary training to develop external verifiers, reward models, or additional joint training for self-verification (Kwok et al., 2025; Nakamoto et al., 2024; Yang et al., 2025; Jang et al., 2025). Such processes necessitate additional data collection and substantial training compute, which limits their scalability and immediate deployment to unseen domains. In contrast, SCALE is entirely training-free and operates in a single inference pass, eliminating both the computational bottleneck of multiple rollouts and the resource-intensive requirement for auxiliary training.

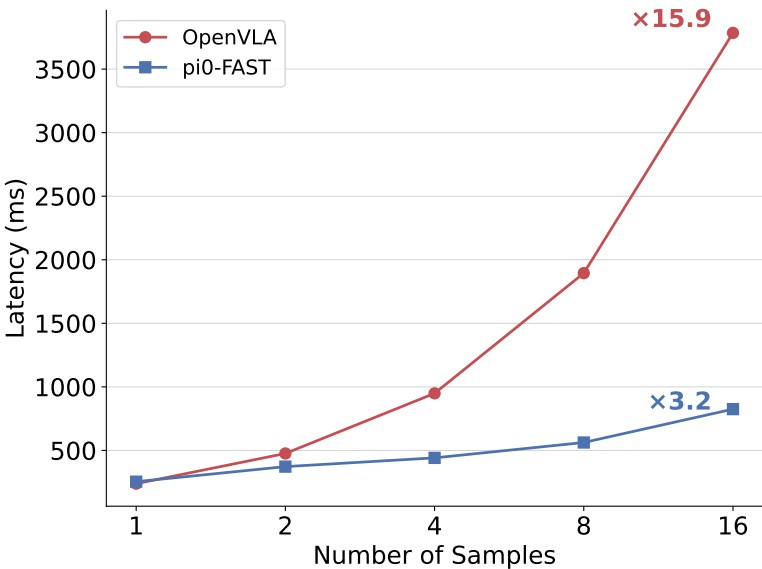

*Figure 6.* **Comparison of Action Generation Latency.** We compare the per-step latency of OpenVLA and $\pi_0$-FAST across varying numbers of generated samples. Latency increases as the number of generated samples grows, with OpenVLA and $\pi_0$-FAST exhibiting $15.9\times$ and $3.2\times$ increases at 16 samples, respectively. SCALE maintains the efficiency of a single inference pass, avoiding the latency costs associated with generating multiple candidates.

## F. Detailed Experimental Setup and Benchmarks

In this section, we provide brief descriptions of the simulation benchmarks and the real-world experimental setup used in our evaluation. Task examples of both simulation and real-world environments are provided in Fig. 8.

### F.1. Simulation Benchmarks

**LIBERO** (Liu et al., 2023) is a widely adopted benchmark for robotic manipulation, designed to evaluate knowledge transfer across diverse distribution shifts. It consists of four distinct task suites—*Spatial*, *Object*, *Goal*, and *Long*—which cover variations in spatial layouts, object types, task objectives, and complex multi-stage manipulation sequences, respectively. Each suite comprises 10 unique tasks, with each task containing 50 episodes.

**SIMPLER-WidowX** (Li et al., 2024) is a *real-to-sim* evaluation framework utilizing the WidowX robot and BridgeData V2 (Walke et al., 2023). It consists of four representative tasks: *Put Spoon on Towel*, *Put Carrot on Plate*, *Stack Green Block on Yellow Block*, and *Put Eggplant in Yellow Basket*, with each task evaluated over 24 episodes. By focusing on these *precise manipulation* tasks with narrow tolerances, the benchmark enables high-fidelity evaluation in simulation environments that closely mirror real-world deployment conditions.

**LIBERO-PRO** (Zhou et al., 2025) evaluates model robustness beyond rote memorization by introducing systematic perturbations to the standard LIBERO suites. It encompasses five dimensions: *Object attribute (Object)*, *Initial position (Swap)*, *Task (Task)*, *Semantic (Language)*, and *Environment (Env)*. Notably, the authors report that even state-of-the-art models suffer significant performance degradation under these perturbations. We focus on the most challenging **LIBERO-PRO-Long** suite to assess whether a model possesses genuine environmental perception or merely relies on memorizing trajectories. We evaluate on the four released perturbation types: Object, Swap, Task, and Language. For an original task such as "turn on the stove and put the moka pot on it," these dimensions manifest as: replacing the moka pot with an *odd moka pot* (**Object**), exchanging the positions of the stove and moka pot (**Swap**), switching the goal to moving a frypan (**Task**), rephrasing the language instruction to "turn stove and place moka pot" (**Language**), or relocating the scene to a living room table (**Env**).

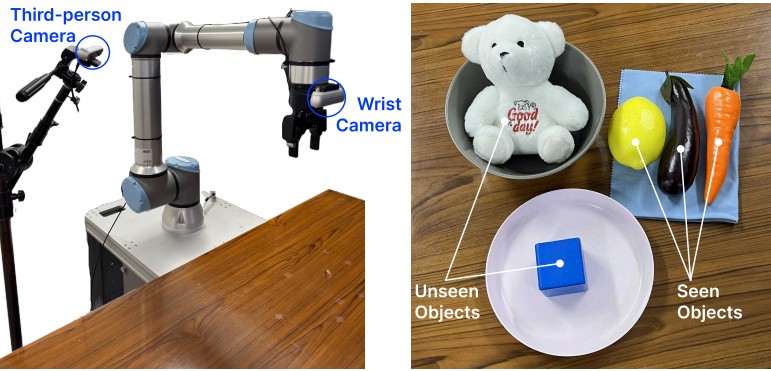

*Figure 7.* (Left) Real-world experimental setup using a UR10e robot with third-person and wrist-mounted cameras. (Right) Examples of seen and unseen objects used in real-world experiments.

### F.2. Real-World Experimental Setup

As shown in Fig. 7, our real-world tabletop experimental setup comprises a UR10e arm equipped with a Robotiq 2F-85 gripper and two Intel RealSense D455 cameras: a third-person camera for global scene understanding and a wrist-mounted camera for localized manipulation. The dataset is collected via teleoperation using Meta Quest 3 (Iyer et al., 2024) at 30Hz.

We consider three *in-distribution* (ID) and two *out-of-distribution* (OOD) pick-and-place tasks. The ID tasks include *Put Carrot on Towel* (**Carrot/Towel**), *Put Eggplant in Bowl* (**Eggplant/Bowl**), and *Put Lemon on Plate* (**Lemon/Plate**). The OOD tasks consist of *Put Teddy Bear in Bowl* (**Teddy Bear/Bowl**) and *Put Cube on Plate* (**Cube/Plate**). Each ID task involves objects with different geometries, whereas the OOD tasks introduce unseen object compliance (a soft teddy bear) and geometry (a small cube), requiring more precise manipulation. Each task consists of 12 distinct episodes with different initial object locations. For the ID tasks, we collect a total of 144 demonstrations, corresponding to 48 demonstrations per task and 4 demonstrations per initial object location. Evaluation is performed over 24 episodes per task, with 2 evaluation episodes for each initial object location.

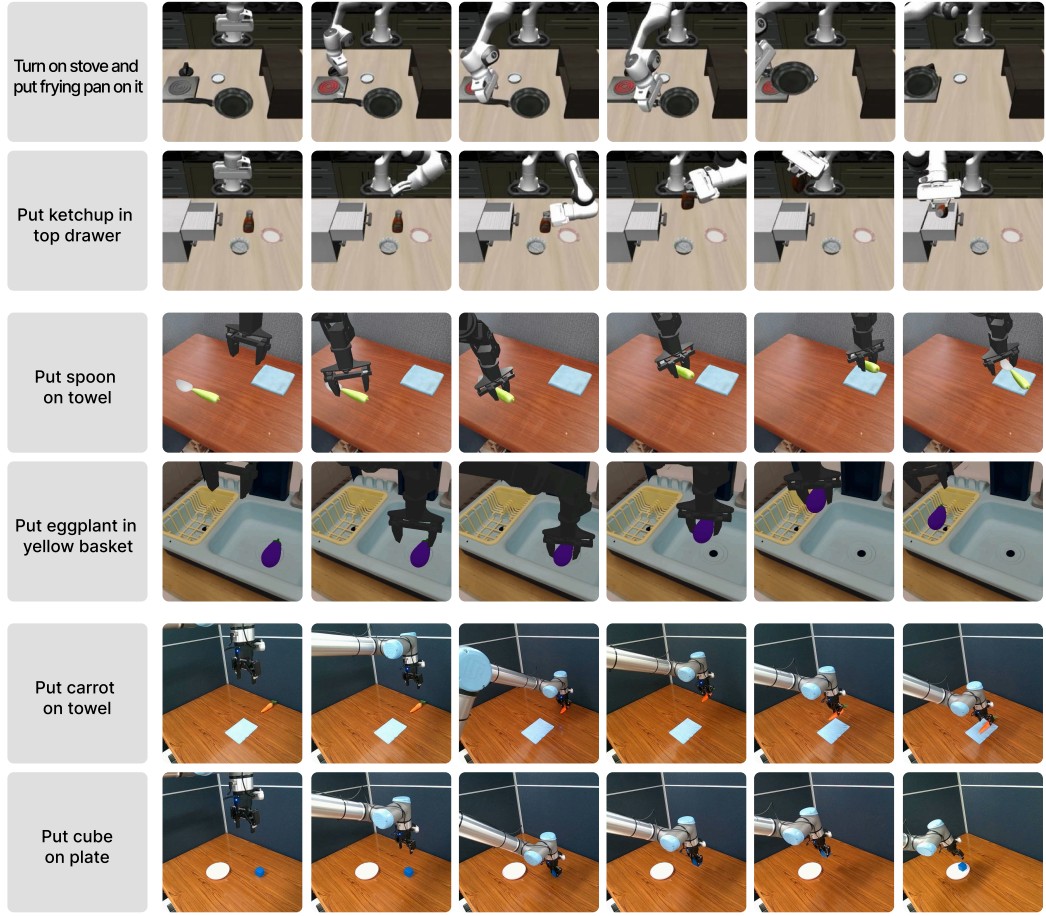

*Figure 8.* Task examples from LIBERO (top two rows), SIMPLER-WidowX (middle two rows), and real-world experiments (bottom two rows).

# G. Implementation Details

In this section, we provide comprehensive details on the architectural distinctions of the VLA backbones, training configurations for simulation and real-world experiments, and specific deployment strategies.

## G.1. Architectural Distinctions across VLA Backbones

While SCALE is backbone-agnostic, its implementation is tailored to the unique vision processing and action tokenization schemes of each model:

- **OpenVLA (Kim et al., 2024):** Built upon a standard VLM paradigm with a fused vision encoder (DINOv2 and SigLIP), this model predicts seven discrete tokens per control step. Each token corresponds to a dimension in the action vector $(x, y, z, \text{roll}, \text{pitch}, \text{yaw}, g)$—representing translation, rotation, and gripper state, respectively—where each dimension is discretized into bins using a unified action token vocabulary $\mathcal{V}$ where $|\mathcal{V}| = 256$.

- $\pi_0$**-FAST (Pertsch et al., 2025):** Utilizes a SigLIP vision tower but diverges in its actuation interface via the FAST tokenizer. FAST applies a Discrete Cosine Transform (DCT) and Byte-Pair Encoding (BPE) to compress action chunks, generating a *variable-length* action token sequence. Actions are recovered by inverting the BPE and DCT operations, with all tokens drawn from a single shared action vocabulary $\mathcal{V}$ where $|\mathcal{V}| = 2048$.

- **SpatialVLA (Qu et al., 2025):** Augments SigLIP features with 3D positional encodings derived from predicted depth. For action tokenization, it compresses each control step into three autoregressive tokens representing translation $(x, y, z)$, rotation $(\text{roll}, \text{pitch}, \text{yaw})$, and gripper state. By employing ensemble-style action chunking with a horizon

of $T = 4$, the model generates a sequence of 12 spatial tokens $(3 \times 4)$ per forward pass. Crucially, it utilizes *factorized* action vocabularies for translation, rotation, and the binary gripper—with sizes of 4096, 4096, and 2, respectively—forming a total action vocabulary $\mathcal{V}$ where $|\mathcal{V}| = 8194$.

## G.2. Training Configurations

### G.2.1. SIMULATION EXPERIMENTS

We evaluate performance on the LIBERO suite (Liu et al., 2023) and SIMPLER-WidowX (Li et al., 2024) for seen tasks, and LIBERO-PRO-Long (Zhou et al., 2025) for unseen generalization.

- **OpenVLA:** We utilize the official model checkpoints provided by the authors, which are fine-tuned for each respective LIBERO task suite. For LIBERO-PRO-Long evaluation, we apply the checkpoints fine-tuned on LIBERO-Long to assess zero-shot robustness.

- $\pi_0$**-FAST:** We perform full fine-tuning on the LIBERO datasets for 10k steps using two NVIDIA H100 GPUs (global batch size 32, action chunk size 10) and evaluate the model on both LIBERO (fine-tuned) and LIBERO-PRO (zero-shot). For SIMPLER-WidowX, the backbone is fine-tuned on BridgeData V2 (Walke et al., 2023) for 10k steps (batch size 64, action chunk size 5).

- **SpatialVLA:** We employ the official checkpoints provided by the authors: `spatialVLA-4B-224-pt` for zero-shot evaluation and `spatialvla-4b-224-sft-bridge` for fine-tuned tasks.

### G.2.2. REAL-WORLD EXPERIMENTS

- **OpenVLA:** Since OpenVLA is pre-trained on BridgeData V2 at 5 Hz, we downsample our dataset to 5 Hz before fine-tuning. We fine-tune the official `openvla-7b` checkpoint using LoRA ($r = 32$) (Hu et al., 2022) on 4 NVIDIA A100 GPUs for 15k steps (batch size 8).

- $\pi_0$**-FAST:** Unlike other VLA models used in simulation experiments and OpenVLA in real-world experiments, $\pi_0$-FAST additionally leverages the robot's proprioceptive state and wrist-mounted camera in real-world experiments to ensure stable execution. We fully fine-tuned the official base $\pi_0$-FAST checkpoint on 4 NVIDIA A100 GPUs for 10k steps (batch size 32). The model predicts a chunk horizon of 20, with the first 15 steps executed during inference.

## G.3. Deployment and Inference

The following sampling strategies and hyperparameter configurations were applied consistently across all simulation benchmarks and real-world experiments to ensure a uniform evaluation of SCALE.

### G.3.1. SAMPLING STRATEGY OF SCALE

To accommodate tokenization variances, we tailor the sampling strategy to the unique properties of each model. For **OpenVLA**, we apply sampling to all 7 action tokens. For $\pi_0$**-FAST**, as the FAST tokenizer generates action tokens in order from low- to high-frequency coefficients, we sample the first 5 tokens corresponding to the primary low-frequency components, following prior work (Jang et al., 2025). For **SpatialVLA**, we sample the first 3 tokens representing the current step. Note that SpatialVLA utilizes factorized vocabularies (Appendix G.1); thus, $u_t^k$ is calculated using the logits corresponding to the respective token type (translation, rotation, or gripper).

Notably, due to the autoregressive nature of these backbones, sampling the initial tokens ensures that the influence of the intervention inherently propagates to all subsequent tokens in the sequence.

### G.3.2. VISUAL ATTENTION MODULATION OF SCALE

While all models utilize a SigLIP encoder, OpenVLA employs a fused DINOv2-SigLIP architecture. We apply visual attention modulation to all vision encoders, including both the DINOv2 and SigLIP in OpenVLA.

### G.3.3. HYPERPARAMETERS

The base temperature $T_0$ is set to 1.0 for OpenVLA and 0.3 for $\pi_0$-FAST and SpatialVLA. For adaptive visual attention, we set $\kappa = 2$, constraining the attention temperature $\gamma_t$ within the interval $(0.5, 2.0)$. The temporal smoothing factor is set to $\alpha = 0.8$ for OpenVLA and SpatialVLA, and $\alpha = 0.66$ for $\pi_0$-FAST to account for its more frequent uncertainty updates in autoregressive chunk generation. We use same hyperparameters for both simulation and real-world experiments.

*Table 10.* Performance comparison of OpenVLA on LIBERO across various decoding strategies and hyperparameters. We evaluate standard sampling, Top-$k$, and Top-$p$ with different settings.

| Method | Spatial | Object | Goal | Long | Avg. |
|---|---|---|---|---|---|
| OpenVLA* (fine-tuned) | 86.2 | 86.2 | 77.7 | 52.7 | 75.7 |
| + Sampling ($t$=0.3) | 85.1 | 87.6 | 79.5 | 53.6 | 76.5 |
| + Sampling ($t$=0.5) | 83.9 | 88.5 | 78.0 | 54.2 | 76.2 |
| + Sampling ($t$=0.7) | 85.2 | 87.6 | 78.9 | 54.4 | 76.5 |
| + Sampling ($t$=1.0) | 85.1 | 87.9 | 78.9 | 54.7 | 76.7 |
| + Top-$k$ ($k$=10, $t$=0.7) | 84.7 | 88.3 | 79.1 | 53.8 | 76.5 |
| + Top-$k$ ($k$=20, $t$=0.7) | 84.7 | 89.0 | 78.2 | 55.0 | 76.7 |
| + Top-$k$ ($k$=40, $t$=0.7) | 85.2 | 88.2 | 78.3 | 55.2 | 76.7 |
| + Top-$k$ ($k$=40, $t$=1.0) | 84.3 | 88.2 | 80.7 | 53.5 | 76.7 |
| + Top-$p$ ($p$=0.9) | 86.9 | 88.1 | 78.6 | 55.1 | 77.2 |
| + Top-$p$ ($p$=0.95) | 85.7 | 88.4 | 77.7 | 55.4 | 76.8 |
| + SCALE (Ours) | **89.5** | **91.0** | **82.3** | **63.3** | **81.5** |

## H. Sensitivity Analysis of Baseline Decoding Strategies

In this section, we provide a detailed sensitivity analysis of standard decoding strategies—temperature sampling, top-$k$ sampling, and top-$p$ sampling—to justify our selection of baseline hyperparameters. We evaluated the OpenVLA model on the LIBERO benchmark across a wide range of hyperparameter configurations. For all baseline strategies, we mask non-action tokens to ensure that sampling is performed only over each model's specific action token vocabulary.

As summarized in Table 10, while various decoding strategies generally improve upon the fine-tuned OpenVLA baseline (75.7% average success rate), the performance gains are relatively marginal and insensitive to specific hyperparameter tuning. For instance, varying the temperature $t$ from 0.3 to 1.0 or adjusting the top-$k$ and top-$p$ thresholds results in average success rates that fluctuate within a narrow range of approximately 76.2% to 77.2%.

SCALE significantly outperforms all fixed-parameter decoding strategies, achieving an average success rate of 81.5%. This result underscores the limitation of fixed hyperparameters: no single manual setting can consistently adapt to the varying levels of predictive uncertainty encountered across different tasks and environmental states. Since the performance of these baseline strategies did not exhibit significant variance across different parameters, we selected the following representative configurations for comparison across all models and benchmarks in our main experiments: temperature sampling with $t$=1.0, top-$k$ sampling with $k$=40 and $t$=0.7, and top-$p$ sampling with $p$=0.9.

## I. Comparison with Test-Time Scaling Methods: Per-Task Breakdown

We provide a detailed comparison between SCALE and existing TTS approaches on LIBERO-Long, a challenging benchmark featuring long-horizon manipulation tasks that require sustained precision over extended episodes. Table 11 reports per-task success rates, where all methods build upon OpenVLA fine-tuned on LIBERO data (OpenVLA*) and apply their respective inference strategies. We compare against two representative TTS methods: RoboMonkey (Kwok et al., 2025), which trains a VLM-based action verifier, and TACO (Yang et al., 2025), which employs a learned value function for action selection—both requiring additional training and multiple forward passes. SCALE achieves the highest average success rate (63.3%), outperforming RoboMonkey (56.5%) by 6.8%p and TACO (60.0%) by 3.3%p, while requiring no additional training and only a single forward pass. Notably, SCALE attains the best performance on 7 out of 10 tasks, with substantial gains on challenging tasks such as "Moka Pots on Stove" (+10.0%p over TACO) and "Mug in Microwave" (+6.0%p over TACO). These results demonstrate that adaptively modulating perception and action based on self-uncertainty can be more effective than selecting from multiple candidates via learned verifiers, particularly for long-horizon tasks where sustained

*Table 11.* Per-task SR (%) on LIBERO-Long with OpenVLA backbone. We compare SCALE against training-free baseline (greedy decoding) and training-required test-time scaling methods. *Reproduced using authors' official implementation.

| Tasks | Training-free, single inference | | Training-required, test-time scaling | |
|---|---|---|---|---|
| | OpenVLA* | SCALE (Ours) | RoboMonkey | TACO |
| Soup and Sauce in Basket | 62.7 | **66.0** | 59.0 | **66.0** |
| Cheese and Butter in Basket | 69.3 | **82.7** | 79.0 | 82.0 |
| Turn on Stove and Place Moka | 58.0 | **58.7** | 58.0 | 52.0 |
| Black Bowl in Drawer | 40.7 | **58.0** | 37.0 | 50.0 |
| Mugs on Plate | 49.3 | 51.3 | **55.0** | 50.0 |
| Book in Caddy | 76.0 | 86.0 | 86.0 | **90.0** |
| Mug and Pudding on Plate | 46.7 | **62.7** | 59.0 | 54.0 |
| Soup and Cheese in Basket | 63.3 | 76.0 | 62.0 | **80.0** |
| Moka Pots on Stove | 20.7 | **38.0** | 26.0 | 28.0 |
| Mug in Microwave | 40.0 | **54.0** | 44.0 | 48.0 |
| Average | 52.7 | **63.3** | 56.5 | 60.0 |

adaptability is crucial.

## J. Details on Baseline Uncertainty Metrics

To ensure a fair comparison with our proposed method, we evaluate several baseline uncertainty metrics: normalized entropy, confidence ($p_{\max}$), Gini impurity (Breiman et al., 1984), and Self-certainty (Kang et al., 2025). We map all metrics to the unit interval $u \in [0, 1]$, where $u = 0$ represents maximum certainty and $u = 1$ represents maximum uncertainty. The specific normalization formulations are defined as follows, given the categorical distribution $p^k$ over the action vocabulary $\mathcal{V}$ at token position $k$:

- **Normalized Entropy:** We use the Shannon entropy normalized by the logarithm of the vocabulary size $|\mathcal{V}|$:

$$u_{\text{ent}}^k = \frac{-\sum_{i \in \mathcal{V}} p_i^k \log p_i^k}{\log |\mathcal{V}|} \tag{18}$$

- **Maximum Probability ($p_{\max}$):** Since $p_{\max}^k = \max_i p_i^k$ corresponds to the model's confidence, we use its complement to represent uncertainty:

$$u_{p_{\max}}^k = 1 - \max_{i \in \mathcal{V}} p_i^k \tag{19}$$

- **Gini Impurity:** We adopt the Gini impurity measure, defined as:

$$u_{\text{gini}}^k = 1 - \sum_{i \in \mathcal{V}} (p_i^k)^2 \tag{20}$$

- **Self-certainty:** Following Kang et al. (2025), we consider the Kullback-Leibler divergence from the uniform distribution $q^{\text{high}}$ to the policy distribution $p^k$, denoted as $D_{\text{KL}}(q^{\text{high}}||p^k)$. To map this measure to $[0, 1]$ while preserving the order of uncertainty, we apply an exponential decay transformation:

$$u_{\text{sc}}^k = \exp(-D_{\text{KL}}(q^{\text{high}}||p^k)) \tag{21}$$

where a high divergence (certain) yields $u_{\text{sc}}^k \approx 0$, and zero divergence (uncertain) yields $u_{\text{sc}}^k = 1$.

**Implementation.** In our comparative experiments, we substitute $\sigma(u^k)$ in Equation 2 directly with these normalized baseline metrics. These surrogate values are used to modulate the action decoding process and visual attention, exactly as described in our proposed method. All other algorithmic components remain identical to the proposed framework, ensuring that the performance differences are attributable solely to the choice of the uncertainty metric.

*Table 12.* **Success rates with standard deviation on LIBERO.** We report the mean and standard deviation over three random seeds for the main LIBERO results using OpenVLA and $\pi_0$-FAST. All values are success rates in percentage. The results show that SCALE consistently improves over fixed decoding baselines across both VLA backbones, with gains that remain larger than the corresponding standard deviations.

| Backbone | Method | Spatial | Object | Goal | Long | Avg. |
|---|---|---|---|---|---|---|
| OpenVLA | OpenVLA* (fine-tuned) | $86.2 \pm 1.1$ | $86.2 \pm 0.7$ | $77.7 \pm 0.7$ | $52.7 \pm 0.9$ | $75.7 \pm 0.6$ |
| | + Sampling ($t$=1.0) | $85.1 \pm 1.2$ | $87.9 \pm 1.0$ | $78.9 \pm 0.7$ | $54.7 \pm 1.2$ | $76.7 \pm 0.7$ |
| | + Top-$k$ ($k$=40, $t$=0.7) | $85.2 \pm 0.9$ | $88.2 \pm 0.2$ | $78.3 \pm 1.3$ | $55.2 \pm 0.7$ | $76.7 \pm 0.4$ |
| | + Top-$p$ ($p$=0.9) | $86.9 \pm 1.5$ | $88.1 \pm 1.3$ | $78.6 \pm 1.3$ | $55.1 \pm 1.0$ | $77.2 \pm 0.8$ |
| | + **SCALE (Ours)** | $\mathbf{89.5 \pm 0.8}$ | $\mathbf{91.0 \pm 0.3}$ | $\mathbf{82.3 \pm 0.6}$ | $\mathbf{63.3 \pm 1.0}$ | $\mathbf{81.5 \pm 0.7}$ |
| $\pi_0$-FAST | $\pi_0$-FAST * (fine-tuned) | $96.6 \pm 0.3$ | $98.1 \pm 0.1$ | $93.7 \pm 0.5$ | $76.3 \pm 0.6$ | $91.2 \pm 0.4$ |
| | + Sampling ($t$=1.0) | $87.0 \pm 1.1$ | $94.6 \pm 0.6$ | $83.5 \pm 0.8$ | $72.2 \pm 1.3$ | $84.3 \pm 0.5$ |
| | + Top-$k$ ($k$=40, $t$=0.7) | $93.7 \pm 0.4$ | $96.5 \pm 0.6$ | $87.5 \pm 1.4$ | $74.8 \pm 0.7$ | $88.1 \pm 0.2$ |
| | + Top-$p$ ($p$=0.9) | $90.2 \pm 1.2$ | $95.3 \pm 0.6$ | $85.9 \pm 0.9$ | $73.4 \pm 1.3$ | $86.2 \pm 0.3$ |
| | + **SCALE (Ours)** | $\mathbf{97.7 \pm 0.3}$ | $\mathbf{98.7 \pm 0.2}$ | $\mathbf{94.7 \pm 0.5}$ | $\mathbf{80.9 \pm 0.4}$ | $\mathbf{93.0 \pm 0.3}$ |

## K. Statistical Reliability of Main Results

As shown in Table 12, SCALE consistently improves over fixed decoding baselines across both OpenVLA and $\pi_0$-FAST. For OpenVLA, SCALE achieves an average success rate of $81.5 \pm 0.7\%$, outperforming the strongest fixed decoding baseline, Top-$p$ sampling, which achieves $77.2 \pm 0.8\%$. On the most challenging LIBERO-Long suite, the gain is even larger: SCALE achieves $63.3 \pm 1.0\%$, compared to $55.2 \pm 0.7\%$ from the strongest baseline.

For $\pi_0$-FAST, fixed sampling-based decoding strategies often degrade performance compared to the fine-tuned baseline, suggesting that naive stochastic decoding can be harmful for this action-tokenization scheme. In contrast, SCALE improves upon the fine-tuned $\pi_0$-FAST baseline across all LIBERO suites, achieving an average success rate of $93.0 \pm 0.3\%$ compared to $91.2 \pm 0.4\%$. These results indicate that the improvements of SCALE are stable across random seeds and are not attributable to a single favorable run.

## L. Hyperparameter Sensitivity Analysis

We analyze the sensitivity of SCALE to its key hyperparameters: the base action decoding temperature $T_0$, the temporal smoothing factor $\alpha$, the visual attention modulation bound $\kappa$, and the reference smoothing parameter $\epsilon$. For each backbone, we vary one hyperparameter at a time while fixing the others to the best-performing values used in the main experiments. All experiments in this section are conducted on the LIBERO-Long benchmark.

Table 13 reports the sensitivity of SCALE to $T_0$, $\alpha$, and $\kappa$ using OpenVLA and $\pi_0$-FAST. Overall, SCALE is robust across a broad range of hyperparameter choices. For both backbones, all tested values of $T_0$ and $\alpha$ consistently outperform the corresponding greedy decoding baselines, i.e., $52.7\%$ for OpenVLA and $76.3\%$ for $\pi_0$-FAST. This suggests that the gains of SCALE are not attributable to a narrowly tuned hyperparameter configuration.

For $T_0$, the optimal value differs across backbones. OpenVLA benefits from a relatively larger temperature, achieving the best performance at $T_0$=1.0, whereas $\pi_0$-FAST performs best at $T_0$=0.3. We attribute this trend to differences in action vocabulary size and action tokenization. $\pi_0$-FAST uses a larger action vocabulary ($|\mathcal{V}|$=2048), where increasing the temperature spreads probability mass over many more candidate tokens and can increase the risk of sampling irrelevant actions. In contrast, OpenVLA uses a smaller action vocabulary ($|\mathcal{V}|$=256), making the candidate action space more constrained and allowing a relatively larger $T_0$ without substantially increasing this risk.

For $\alpha$, performance remains stable across the tested range for both backbones. Although the best value differs slightly by backbone, all tested values improve over the respective greedy decoding baselines. This indicates that the temporal smoothing mechanism is not highly sensitive to a specific choice of $\alpha$.

Compared to $T_0$ and $\alpha$, $\kappa$ is more sensitive. For OpenVLA, increasing $\kappa$ to 3.0 causes a substantial performance drop, from $63.3\%$ at $\kappa$=2.0 to $39.8\%$. A similar trend is observed for $\pi_0$-FAST, where performance decreases from $80.9\%$ at $\kappa$=2.0 to $74.4\%$ at $\kappa$=3.0. This is expected because $\kappa$ directly controls the range of visual attention temperature, i.e., $\gamma_t \in (1/\kappa, \kappa)$.

*Table 13.* **Hyperparameter sensitivity analysis of SCALE on LIBERO-Long.** We report success rates (SR, %) for OpenVLA and $\pi_0$-FAST while varying one hyperparameter at a time and fixing the others to their best-performing values.

|  | OpenVLA | | | | $\pi_0$-FAST | | | |
|---|---|---|---|---|---|---|---|---|
| $T_0$ | 0.3 | 0.5 | 0.7 | 1.0 | 0.3 | 0.5 | 0.7 | 1.0 |
| SR (%) | 57.0 | 56.6 | 59.4 | **63.3** | **80.9** | 77.2 | 76.4 | 77.0 |
| $\alpha$ | 0.66 | 0.75 | 0.8 | 0.9 | 0.66 | 0.75 | 0.8 | 0.9 |
| SR (%) | 60.4 | 62.4 | **63.3** | 61.2 | **80.9** | 80.0 | 79.0 | 79.8 |
| $\kappa$ | 1.5 | 2.0 | 2.5 | 3.0 | 1.5 | 2.0 | 2.5 | 3.0 |
| SR (%) | 59.8 | **63.3** | 56.8 | 39.8 | 79.2 | **80.9** | 77.8 | 74.4 |

*Table 14.* **Sensitivity of success rate to $\epsilon$.** We report success rates on LIBERO-Long using OpenVLA with only the adaptive action decoding component of SCALE. Performance remains stable across small magnitudes of $\epsilon$.

| Method | SR (%) |
|---|---|
| OpenVLA* (fine-tuned) | 52.7 |
| + Ada. Decoding ($\epsilon=10^{-10}$) | 57.2 |
| + Ada. Decoding ($\epsilon=10^{-11}$) | 57.4 |
| + Ada. Decoding ($\epsilon=10^{-12}$) | **58.0** |
| + Ada. Decoding ($\epsilon=10^{-13}$) | 57.0 |
| + Ada. Decoding ($\epsilon=10^{-14}$) | 57.6 |

An excessively large $\kappa$ can over-flatten visual attention, causing the vision encoder to spread attention too broadly and weaken task-relevant visual grounding. Nevertheless, within a reasonable range, $\kappa \in [1.5, 2.5]$, SCALE still outperforms the greedy decoding baseline for both backbones.

Table 14 reports the sensitivity to $\epsilon$, which determines the probability mass assigned to non-top-1 tokens in the low-uncertainty reference distribution $q^{\text{low}}$. To isolate the effect of $\epsilon$, we evaluate OpenVLA with only the adaptive action decoding component of SCALE. Varying $\epsilon$ over $\{10^{-10}, 10^{-11}, 10^{-12}, 10^{-13}, 10^{-14}\}$ yields success rates between $57.0\%$ and $58.0\%$, with all tested values outperforming the greedy decoding baseline by at least $4.3$ percentage points. This suggests that as long as $\epsilon$ is sufficiently small to preserve the near-deterministic nature of $q^{\text{low}}$, the exact numerical value has only a marginal effect on performance.

Finally, we emphasize that hyperparameters are selected using only LIBERO-Long for each backbone. The same hyper-parameter configuration is then applied to all other simulation benchmarks and real-world experiments without further benchmark-specific adjustment. Despite this, SCALE consistently improves performance across the main experimental results, indicating that the method does not rely on rigid per-benchmark manual tuning.

