# OpenReview forum: "SCALE: Self-uncertainty Conditioned Adaptive Looking and Execution for Vision-Language-Action Models"
_ICML.cc/2026/Conference — ICML 2026 spotlight_

### Official Review · Reviewer_hsvz · 2026-03-11

**Soundness:** 4
**Presentation:** 3
**Significance:** 4
**Originality:** 4
**Overall Recommendation:** 5
**Confidence:** 4

**Summary:**

The paper proposes SCALE, a method that leverages a VLA's own uncertainty (defined with a novel metric) to adapt both the action decoding temperature and the visual attention temperature at test time. Uncertainty is defined as the difference in KL divergence of the model's distribution from a maximally uncertain (flat uniform) distribution, and from a maximally certain (spike on top 1) distribution, which is equivalent to the expectation of the log ratio of the two distributions under the current model. With this, action temperature is chosen to be high under high uncertainty (explore more actions) and low under low uncertainty (focus on confident actions). Similarly, a temperature is added to the denominator of the visual attention layers, which makes attention spread more under uncertainty (include more objects) and lower under certainty (just look at what's most important). The method is tested comprehensively on three simulated benchmarks and in a real world transfer experiment, showing consistently improved performance, with ablations supporting some design choices.

**Compliance With Llm Reviewing Policy:**

Affirmed.

**Final Justification:**

My concerns have been adequately addressed in the rebuttal. I confirm the positive score.

**Key Questions For Authors:**

- Table 8 discusses the choice of where to modulate visual attention, either in the unimodal or cross modal blocks. While the authors make a good case for why unimodal attention is better (modulating what to integrate, not how it's integrated), I wonder if modulating both together would improve the results. As far as I can tell, only either one in isolation was tested.
- Are there some failure cases of the method that the authors observed that would be relevant to show? For example, broadening visual attention too far seems like it would break the method. T0 changes a lot across baselines, was this because of some particular failures?
-  In general, how were the hyperparameters set?

**Limitations:**

Limitations of the method are not explicitly discussed in a dedicated paragraph, just at times during the paper (for example, when talking about the assumption of consecutive frames being similar). It would be goo to have a paragraph discussing limitations and future directions. Impact Statement is sufficiently detailed.

**Strengths And Weaknesses:**

### Strengths
- The introduced uncertainty measure is very simple and elegant, combining previous ideas (e.g., entropy which mostly measures against a flat distribution, or looking at only the top1 token) and statistics with the log-likelihood test analogy. The motivation is very clear and its design is supported by the experiment in Table 7.
- Modulating not just the attention decoding temperature, but also the visual attention temperature, is very interesting. I appreciate the fact that the method is general, so that it could be applied to any VLA (as long as it's visual encoder is based on attention) without any training. This makes it very applicable to many potential settings.
- The experiments have good coverage on LIBERO and Simpler, and different VLA backbones being tested. The results are good, with some tasks benefiting more than others. On average I'd say it looks like gains are more pronounced on less saturated tasks (with some exceptions), which is a good sign. Real world results are also promising.
- The ablation experiments in Tables 6-8 are very welcome and validate the key design choices of the method, namely the uncertainty measure, decoding strategies and visual attention modulation. The results on all components are convincing. I also appreciate the qualitative analysis showing some examples of the model's behavior. I particularly like seeing the effect of visual attention.

### Weaknesses
- The method introduces many hyperparameters, but the paper doesn't discuss their impact on the method's performance, besides epsilon in Appendix B. From Appendix H, it seems like the values of T0, kappa and alpha were changed depending on the VLA model. While this is fair, it also suggests that the method might be sensitive to these, so a discussion on how much this impacts performance and how these parameters were chosen would be beneficial.
- A key assumption made is that the difference in consecutive frames is minimal, which avoids the need for another forward pass. This seems to be fair enough in the settings that were tested (which are mostly static manipulation), but this might not always be the case. For instance, in a dynamic setting with many moving objects, SCALE could need an extra forward pass in order to provide benefits. This would be interesting to test.
- Minor, but I feel like Figure 2 is very busy: it explains 4 concepts together and is very dense in arrows and notation, which makes is pretty hard to follow. During my first read, I ended up skipping it and returning after I finished reading, so I would say it didn't help to understand the method. I would suggest cleaning it up and maybe even dividing it into cleaner self contained single-column figures with more visual intuition and less notation.

---

> ### Author Rebuttal · Authors · 2026-03-31
>
> > (**W1**, **Q3**): SCALE might be sensitive to hyperparameters and how were the hyperparameters set.
>
> $\to$ We provide sensitivity analysis in Table M, where all tested $T_0$ and $\alpha$ consistently outperform the baselines. $T_0$ shows divergent trends across backbones due to action vocabulary size differences, while $\alpha$ remains stable across the tested range. Although high $\kappa$ can cause degradation, SCALE outperforms the baseline across a broad range. Notably, the best configuration from LIBERO-Long was fixed across all other benchmarks and real-world experiments without further tuning.
>
> \begin{array}{l|cccc|cccc} \hline
>  &  \rlap{~~~~~~~~~\text{OpenVLA}} & & & & \rlap{~~~~~~~~~\text{$\pi_0$-FAST}}
> \newline \hline T_0 & 0.3 & 0.5 & 0.7 & 1.0 & 0.3 & 0.5 & 0.7 & 1.0
> \newline \text{SR (\\%)} & 57.0 & 56.6 & 59.4 & \mathbf{63.3} & \mathbf{80.9} & 77.2 & 76.4 & 77.0
> \newline \hline \alpha & 0.66 & 0.75 & 0.8 & 0.9 & 0.66 & 0.75 & 0.8 & 0.9
> \newline \text{SR (\\%)} & 60.4 & 62.4 & \mathbf{63.3} & 61.2 & \mathbf{80.9} & 80.0 & 79.0 & 79.8
> \newline \hline \kappa & 1.5 & 2.0 & 2.5 & 3.0 & 1.5 & 2.0 & 2.5 & 3.0
> \newline \text{SR (\\%)} & 59.8 & \mathbf{63.3} & 56.8 & 39.8 & 79.2 & \mathbf{80.9} & 77.8 & 74.4
> \newline \hline \end{array}
>
> **Table M.** Hyperparameter sensitivity analysis of SCALE.
>
> ---
>
> > (**W2**): In a dynamic setting with many moving objects, SCALE could need an extra forward pass in order to provide benefits. This would be interesting to test.
>
> $\to$ We conduct additional *dynamic* experiments using $\pi_0$-FAST [1] in both simulation and real-world settings. We evaluate on the dynamic DOMINO benchmark [2] across four tasks (100 episodes per task). As shown in Table N, SCALE consistently improves over greedy decoding, and the Two-step Oracle achieves a further gain. We observe a consistent pattern in real-world OOD experiments as shown in Table O: on a dynamic pick task where the target object (lemon) moves at approximately 5cm/s during execution (25 episodes each), SCALE provides meaningful improvements.
> When we further increased the object speed, all methods achieved zero success rate. Indeed, handling highly dynamic scenarios remains an open challenge for current VLA architectures—both OpenVLA [3] and $\pi_0$-FAST [1] identify this limitation.
> | Method | shake bottle | shake bottle horizontally | adjust bottle | place container plate | Avg. |
> | :--- | :---: | :---: | :---: | :---: | :---: |
> | $\pi_0$-FAST (greedy) | 27 | 28 | 11 | 13 | 19.8 |
> | + SCALE (**ours**) | 35 | 35 | 19 | 17 | 26.5 |
> | + Two-Step Oracle | 38 | 37 | 21 | 22 | 29.5 |
>
> **Table N.** Success rates (%) on dynamic DOMINO [2] simulation tasks.
>
> | Method | Moving object (5 cm/s) | Moving object (10 cm/s) |
> | :--- | :---: | :---: |
> | $\pi_0$-FAST (greedy) | 52 | 0 |
> |  + SCALE (**ours**) | 68 | 0 |
> |  + Two-Step Oracle | 72 | 0 |
>
> **Table O.** Success rates (%) on real-world dynamic task at varying speeds.
>
> [1] Pertsch et al., Fast: Efficient Action…, RSS, 2025.\
> [2] Fang et al., Towards Generalizable…, arXiv, 2026. \
> [3] Kim et al., OpenVLA: An Open-Source…, CoRL, 2024.
>
> ---
>
> > (**Q1**): I wonder if modulating both visual attention (unimodal and cross modal) would improve the results.
>
> $\to$ As shown in Table P, modulating both together yields 60.8%, which is better than modulating $\pi_\theta$ cross-modal attention alone but lower than modulating $f_\phi$ uni-modal attention alone. We conjecture that modulating cross-modal attention on top of already-modulated visual features introduces redundant or conflicting adjustments. Notably, combining both modulations improves over cross-modal attention alone, confirming the importance of uni-modal visual attention modulation.
>
> | Modulation Target | Success Rate on LIBERO-Long (%) |
> | :--- | :---: |
> | $\pi_{\theta}$ cross-modal attn. | 57.4 |
> | $f_{\phi}$ uni-modal attn. | **63.3** |
> | both $\pi_{\theta}$ and $f_{\phi}$ | 60.8 |
>
> **Table P**. Performance comparison of attention modulation targets with OpenVLA.
>
> ---
>
> > (**Q2**): Are there some failure cases of the method that the authors observed that would be relevant to show?
>
> $\to$ SCALE's effectiveness is inherently dependent on the base model's capability. The Swap perturbation in Table 4 exemplifies this: when the base model achieves near-zero performance, no inference-time strategy can compensate for a fundamental lack of task-relevant capability. The reviewer's intuition is correct---broadening attention too far can hurt performance, and as shown in W1, $\kappa$ is more sensitive than other hyperparameters. We also found that $T_0$ should be adjusted per backbone: larger action vocabularies require lower $T_0$ to avoid sampling from the broader tail, while smaller vocabularies allow relatively higher $T_0$.

---

> > ### Author Rebuttal · Reviewer_hsvz · 2026-04-04
> >
> > My concerns have been adequately addressed. I wll keep the positive score.

---

> > > ### Author Response · Authors · 2026-04-04
> > >
> > > We thank Reviewer **hsvz** for the time you have dedicated to reviewing our work. We are pleased that our rebuttal has addressed your concerns adequately.
> > >
> > > If you have any further questions about the paper, please feel free to ask; we would be happy to answer them.
> > >
> > > Best regards, \
> > > Authors

---

### Official Review · Reviewer_pC2n · 2026-03-13

**Soundness:** 3
**Presentation:** 3
**Significance:** 3
**Originality:** 2
**Overall Recommendation:** 4
**Confidence:** 3

**Summary:**

The paper proposes SCALE, a test-time inference strategy for Vision-Language-Action (VLA) models. It calculates a self-uncertainty score from the output logits without requiring extra training or external verifiers. This score dynamically adjusts both the action decoding temperature and the visual attention temperature in a single forward pass. The authors evaluate SCALE on simulation benchmarks like LIBERO and SIMPLER-WidowX, as well as on a real-world robot setup. The results show improvements over greedy decoding and other fixed sampling baselines.

**Compliance With Llm Reviewing Policy:**

Affirmed.

**Key Questions For Authors:**

1. Can you clarify exactly which baseline results were reproduced in your environment and which were copied from prior papers?

2. How often does the delayed visual attention modulation cause issues or overshooting when the environment changes suddenly?

3. How sensitive is the method to the EMA hyperparameter when running at different control frequencies?

4. Please reproduce at least one strong test-time scaling baseline (like RoboMonkey or TACO) in your exact evaluation pipeline to ensure a fair comparison.

5. Provide a deeper failure case analysis for the LIBERO-PRO-Long Task Swap benchmark to explain the limitations of the current method

**Limitations:**

ye s

**Strengths And Weaknesses:**

Strengths

1. The problem is highly relevant. Making VLA models more robust at test time without the heavy cost of multiple rollouts or training extra verifiers is a very practical goal.

2. The proposed method is elegant and easy to deploy. It cleverly reuses the model's own output distribution to guide both perception and action within a single forward pass.

3. The paper provides a good breadth of experiments. It tests the method across different VLA architectures like OpenVLA, FAST, and SpatialVLA in both simulation and real-world settings.

4. The ablation studies clearly show that adaptive visual attention and adaptive action decoding complement each other well to achieve the best performance.

Weakness

1. The comparison with existing test-time scaling methods is not entirely fair. In Table 1, the results for RoboMonkey, TACO, and MG-Select are taken directly from their respective papers, while the proposed method is evaluated locally. This makes it hard to tell if the improvement comes from the method itself or from different evaluation setups.

2. The paper lacks sufficient statistical evidence. The simulation results use only three seeds, and the real-world experiments use only 24 episodes per task. There are no standard deviations or confidence intervals reported in the main tables to prove the results are statistically significant.

3. The real-world evaluation is somewhat limited in scope. It only includes a few simple pick-and-place tasks with models fine-tuned on 48 demonstrations per task. It does not show how the method handles more complex environmental variations like clutter or lighting changes.

4. The paper does not deeply analyze failure cases. For example, the method barely improves performance on the Task Swap perturbation in LIBERO-PRO-Long, but there is little discussion on why it fails there.

---

> ### Author Rebuttal · Authors · 2026-03-31
>
> > (**W1**): The comparison with existing test-time scaling methods is not entirely fair.
>
> $\to$ We argue the comparison is fair (Table below): our reproduced OpenVLA matches the original, and TACO closely matches its reported performance on our setup. For MG-Select and RoboMonkey, we report their published numbers as code is unavailable. On LIBERO-Long, SCALE achieves a larger improvement over all baselines (+6.0%).
>
> | Method | Source | Spatial | Object | Goal | Long | Avg. |
> | :--- | :--- | :---: | :---: | :---: | :---: | :---: |
> | OpenVLA | OpenVLA [1] | 84.7 | 88.4 | 79.2 | 53.7 | 76.5 |
> | OpenVLA | Reproduced | 86.2 | 86.2 | 77.7 | 52.7 | 75.7 |
> | + SCALE (**Ours**) | Reproduced | **89.5** | **91.0** | **82.3** | **63.3** | **81.5** |
> | TACO | Reproduced | - | - | - | 59.8 | - |
> | OpenVLA | MG-Select [2] | 85.2 | 63.7 | 75.5 | 52.5 | 69.2 |
> | MG-Select | MG-Select [2] | 81.7 | 72.5 | 73.6 | 55.4 | 70.8 |
> | OpenVLA | TACO [3] | - | - | - | 54.0 | - |
> | TACO | TACO [3] | - | - | - | 60.0 | - |
> | OpenVLA | RoboMonkey [4] | - | - | - | 49.8 | - |
> | RoboMonkey | RoboMonkey [4] | - | - | - | 56.5 | - |
>
> [1] Kim et al., OpenVLA: An Open-Source…, CoRL, 2024.\
> [2] Jang et al., Verifier-free Test-Time…, arXiv, 2025.\
> [3] Yang et al., Steering Vision-Language-Action…, arXiv, 2025.\
> [4] Kwok et al., RoboMonkey: Scaling Test-Time…, CoRL, 2025.
>
> ---
>
> > (**W2**): The paper lacks sufficient statistical evidence. The simulation results use only three seeds, and the real-world experiments use only 24 episodes per task.
>
> $\to$ We follow standard evaluation protocols from prior VLA work: 3 random seeds in simulation [1--6] and 24 episodes per task in real-world experiments, on par with or above the standard [1--4]. We will provide standard deviations across 3 seeds in the revision.
>
> [1] Kim et al., OpenVLA: An Open-Source…, CoRL, 2024. \
> [2] Qu et al., SpatialVLA: Exploring…, RSS, 2025. \
> [3] Zhao et al., CoT-VLA: Visual Chain-…, CVPR, 2025. \
> [4] Jang et al., Verifier-free Test-Time…, arXiv, 2025.\
> [5] Yang et al., Vision-Language-Action Instruction Tuning:..., ICLR, 2026. \
> [6] Kwok et al., RoboMonkey: Scaling Test-Time…, CoRL, 2025.
>
> ---
>
> > (**W3**): The real-world eval. is limited (pick-and-place tasks). How does SCALE handle  complex variations, clutter or lighting changes?
>
> $\to$ Our 48 demonstrations per task is within the prior VLA work [1,2,3,4]. To address environmental complexity, we conduct additional real-world experiments using $\pi_0$-FAST [5] following the LIBERO-Plus [6] perturb protocol, evaluated zero-shot (no fine-tuning) under cluttered scenes and lighting variations (36 episodes each). SCALE consistently improves over greedy decoding (Clutter: 36.1\%\$\to$52.8\%; Lighting: 72.2\% $\to$86.1\%), confirming its effectiveness under complex environmental variations.
>
> [1] Lin et al., OneTwoVLA: A Unified Vi…, ICLR, 2026. \
> [2] Koo et al., HAMLET…, ICLR, 2026. \
> [3] Kim et al., OpenVLA: An..., CoRL, 2024. \
> [4] Li et al., 3DS-VLA: A 3D…, CoRL, 2025. \
> [5] Pertsch et al., Fast: Effi…, RSS, 2025.\
> [6] Fei et al., LIBERO-Plus…, arXiv, 2025.
>
> ---
>
> > (**W4**, **Q5**): The paper does not deeply analyze failure cases, provide a deeper failure case analysis.
>
> $\to$ The Swap perturbation is a failure case inherent to the base model's capability, not SCALE specifically. With OpenVLA, all methods achieve zero success (Table 4), including the training-required TACO.
>
> ---
>
> > (**Q1**): Can you clarify exactly which baseline results were reproduced in your environment and which were copied from prior papers?
>
> $\to$ Only three results in Table 1 (MG-Select, TACO, RoboMonkey) are copied from their respective papers; all other results across Tables 1–8 are reproduced using each method’s official implementation, as noted in the caption of each table.
>
> ---
>
> > (**Q2**): How does SCALE's one-step lag affect perf. in dynamics?
>
> $\to$ For this, we conduct dynamic experiments. Due to space constraints, we refer the reviewer to our response to Reviewer PHyX(W3), NEyT(W2, Q2) for the detailed results and discussion.
>
> ---
>
> > (**Q3**): Sensitivity of $\alpha$ with different control frequencies
>
> $\to$ We evaluate $\pi_0$-FAST on LIBERO-Long while varying control frequency and $\alpha$. As shown in Table, higher frequencies favor larger ones, while lower frequencies favor smaller ones.
>
> | Control f. | $\alpha=0.66$ | $\alpha=0.75$ | $\alpha=0.8$ | $\alpha=0.9$ | Greedy |
> | :--- | :---: | :---: | :---: | :---: | :---: |
> | 2Hz | **80.9** | 80.0 | 79.0 | 79.8 | 76.3 |
> | 3.3Hz | 68.6 | 68.8 | **71.8** | 71.0 | 64.8 |
> | 5Hz | 64.4 | 63.8 | 65.2 | **65.8** | 61.2 |
>
> ---
>
> > (**Q4**): reproduce at least one strong TTS baseline.
>
> $\to$ We reproduce TACO, using its publicly available evaluation code.
>
> | Method | Source | Spatial | Object | Goal | Long |
> | :--- | :--- | :---: | :---: | :---: | :---: |
> | TACO | Reproduced | - | - | - | 59.8 |
> | TACO | Reported | - | - | - | 60.0 |

---

> > ### Author Rebuttal · Reviewer_pC2n · 2026-04-03
> >
> > I believe 4 is a fair rate for this work. My concerns are partially resolved .

---

> > > ### Author Response · Authors · 2026-04-03
> > >
> > > Dear Reviewer **pC2n**, \
> > > Thank you for your response and for the time you have dedicated to reviewing our work.
> > >
> > > We are glad to hear that your concerns are partially resolved. Furthermore, if you could specify the remaining points, we would be happy to provide further clarification, as we believe the review process is a valuable opportunity to collaboratively resolve ambiguities.
> > >
> > > Best regards, \
> > > The Authors

---

### Official Review · Reviewer_NEyT · 2026-03-13

**Soundness:** 2
**Presentation:** 2
**Significance:** 3
**Originality:** 2
**Overall Recommendation:** 4
**Confidence:** 4

**Summary:**

The paper proposes SCALE, a training-free, single-pass inference strategy for VLA models that adaptively modulates both action decoding and visual perception based on a self-uncertainty signal. The core idea is a dual-reference uncertainty measure which evaluates the model’s predicted token distribution against a one-hot "full certainty" distribution and a uniform "full ambiguity" distribution. This metric is used to dynamically gate the action-token temperature at each decoding step and modulate the vision encoder’s attention temperature across timesteps based on deviations from a recent uncertainty EMA.

**Compliance With Llm Reviewing Policy:**

Affirmed.

**Final Justification:**

The authors have fully addressed my concerns and provided additional experiments to demonstrate the effectiveness of the proposed method. I updated my score to 4.

**Key Questions For Authors:**

Q1. "confidently wrong" cases: In out-of-distribution scenarios where the VLA model hallucinates but remains highly confident, SCALE will blindly perform the wrong operations.

Q2. SCALE uses the uncertainty deviation from the previous timestep to modulate visual processing at the current timestep. How much performance degradation would this cause in dynamic environments where objects are moving, falling, or undergoing active human intervention?

**Limitations:**

The authors should discuss the limitations of this work, such as the sensitivity of hyperparameters and robustness in dynamic environments.

**Strengths And Weaknesses:**

## Strength
S1. The introduction of a self-uncertainty metric based on the KL divergence relative to dual references is both simple and theoretically grounded, effectively yielding an expected log-likelihood ratio.

S2. The authors provide extensive evaluations across three VLA backbones in diverse settings (both simulation and real-world), demonstrating consistent performance improvements.

## Weakness
W1. While the application to VLAs is interesting, the core mechanisms (temperature scaling, EMA smoothing, and distribution-based uncertainty estimation) are relatively standard. The novelty of this work is limited.

W2. SCALE relies on the uncertainty deviation from the previous timestep to adjust the vision encoder at the current timestep. In dynamic environments, this one-step lag could cause the robot's vision encoder to process critical, highly anomalous frames with outdated attention parameters.

W3. Hyperparameter Sensitivity: The method introduces several new hyperparameters that dictate the delicate balance between exploration and exploitation. These parameters require rigid manual tuning depending on the underlying model architecture.

W4. The real-world hardware experiments are somewhat limited in complexity (e.g., primarily static "Put A on B" tasks). The system's robustness remains unvalidated in dynamic environments and mobile manipulation scenarios, where uncertainty is likely to fluctuate unpredictably.

---

> ### Author Rebuttal · Authors · 2026-03-31
>
> > (**W1**): The novelty of this work is limited.
>
> $\to$ While the individual components are standard, our contributions are: (1) a novel dual-reference uncertainty $u^k_t$ grounded in log-likelihood ratio testing [1,2], outperforming all alternatives by at least 5.5pp (Table 7); and (2) joint perception-action modulation where $u^k_t$ drives both visual attention and action decoding, with EMA-based deviation outperforming instantaneous uncertainty (Table 8; 63.3\% *vs.* 55.4\%). To our knowledge, SCALE is the first training-free method for joint perception-action modulation in VLAs---recognized as novel (PHyX), elegant (hsvz), and practically relevant (pC2n).
>
> [1] Neyman & Pearson, On the problem…, Phil. Trans. R. Soc., 1933.\
> [2] Kullback & Leibler, On information and sufficiency, Ann. Math. Statist., 1951.
>
> ---
>
> > (**W2,Q2**): How does SCALE's one-step lag in uncertainty deviation affect performance in dynamic environments?
>
> $\to$ In our standard settings, consecutive frames are highly correlated, making the previous step's uncertainty a reliable proxy (Tables 1--5). For dynamic environments, we conduct additional experiments using $\pi_0$-FAST [2]: in simulation (DOMINO [3], 4 tasks, 100 episodes each, Table C), SCALE improves over greedy decoding (19.8\%$\to$26.5\%), and the Two-step Oracle (using current attn. signal in Appendix E) achieves 29.5\%; in real-world OOD dynamic pick tasks (5cm/s, 25 episodes, Table D), SCALE improves (52\%$\to$68\%), with the Two-step Oracle reaching 72\%. At higher speed (10cm/s), all methods achieve zero success---a limitation inherited from the base VLA [1,2].
>
> | Method | shake bottle | shake bottle horizontally | adjust bottle | place container plate | Avg. |
> | :--- | :---: | :---: | :---: | :---: | :---: |
> | $\pi_0$-FAST (greedy) | 27 | 28 | 11 | 13 | 19.8 |
> | + SCALE (**ours**) | 35 | 35 | 19 | 17 | 26.5 |
> | + Two-Step Oracle | 38 | 37 | 21 | 22 | 29.5 |
>
> **Table C.** Success rates (%) on dynamic DOMINO.
>
> | Method | Moving object (5 cm/s) | Moving object (10 cm/s) |
> | :--- | :---: | :---: |
> | $\pi_0$-FAST (greedy) | 52 | 0 |
> |  + SCALE (**ours**) | 68 | 0 |
> |  + Two-Step Oracle | 72 | 0 |
>
> **Table D.** SR (%) on real-world task (“Pick up the lemon”).
>
>
> [1] Kim et al., OpenVLA: An Open-Source…, CoRL, 2024. \
> [2] Pertsch et al., Fast: Efficient Action…, RSS, 2025.\
> [3] Fang et al., Towards Generalizable…, arXiv, 2026.
>
> ---
>
> > (**W3**): Hyperparameter Sensitivity.
>
> $\to$ As shown in Table E (and Table 9), all tested values of $T_0$, $\alpha$, and $\epsilon$ consistently outperform greedy decoding across both backbones. $\kappa$ is more sensitive as it directly bounds visual attention temperature, though within $\kappa \in [1.5, 2.5]$ all values still exceed the baseline. Notably, hyperparameters are tuned only on LIBERO-Long and applied without adjustment across all other benchmarks and real-world experiments (Tables 1--5).
>
> \begin{array}{l|cccc|cccc} \hline
>  &  \rlap{~~~~~~~~~\text{OpenVLA}} & & & & \rlap{~~~~~~~~~\text{$\pi_0$-FAST}}
> \newline \hline T_0 & 0.3 & 0.5 & 0.7 & 1.0 & 0.3 & 0.5 & 0.7 & 1.0
> \newline \text{SR (\\%)} & 57.0 & 56.6 & 59.4 & \mathbf{63.3} & \mathbf{80.9} & 77.2 & 76.4 & 77.0
> \newline \hline \alpha & 0.66 & 0.75 & 0.8 & 0.9 & 0.66 & 0.75 & 0.8 & 0.9
> \newline \text{SR (\\%)} & 60.4 & 62.4 & \mathbf{63.3} & 61.2 & \mathbf{80.9} & 80.0 & 79.0 & 79.8
> \newline \hline \kappa & 1.5 & 2.0 & 2.5 & 3.0 & 1.5 & 2.0 & 2.5 & 3.0
> \newline \text{SR (\\%)} & 59.8 & \mathbf{63.3} & 56.8 & 39.8 & 79.2 & \mathbf{80.9} & 77.8 & 74.4
> \newline \hline \end{array}
>
> **Table E.** Hyperparameter sensitivity analysis of SCALE
>
> ---
>
> > (**W4**): The real-world experiments are somewhat limited in complexity.
>
> $\to$ Our real-world evaluation focuses on pick-and-place tasks, following the standard protocol of prior VLA work [1--3] and more. To address concerns about robustness, we conduct additional real-world experiments following the perturbation protocol of LIBERO-Plus [4], evaluating $\pi_0$-FAST zero-shot (no fine-tuning) under cluttered scenes and lighting variations (36 episodes each): SCALE consistently improves over greedy decoding (Clutter: 36.1\%$\to$52.8\%; Lighting: 72.2\%$\to$86.1\%), confirming its effectiveness beyond standard settings.
>
> [1] Jang et al., Verifier-free Test-Time…, arXiv, 2025.\
> [2] Kwok et al., RoboMonkey: Scaling Test-Time…, CoRL, 2025.\
> [3] Peng et al., DAM-VLA: A Dynamic…, arXiv, 2026.\
> [4] Fei et al., LIBERO-Plus: In-depth…, arXiv, 2025.
>
> ---
>
> > (**Q1**): "confidently wrong" cases
>
> $\to$ Confident-but-incorrect predictions can occur in OOD scenarios [1,2]. However, even when $u^k_t$ remains low, SCALE's non-zero sampling temperature still allows exploration beyond the top-1 token, unlike greedy decoding---reflected in consistent gains in unseen simulation (Table 4) and OOD real-world experiments (Table 5).
>
> [1] Guo et al., On Calibration of Modern…, ICML, 2017.\
> [2] Zollo & Zemel, Confidence Calibration…, arXiv, 2025.

---

> > ### Author Rebuttal · Reviewer_NEyT · 2026-04-04
> >
> > The authors have fully addressed my concerns and provided additional experiments to demonstrate the effectiveness of the proposed method. I updated my score to 4.

---

> > > ### Author Response · Authors · 2026-04-04
> > >
> > > We thank Reviewer **NEyT** for the reply.
> > >
> > > To your reply, one thing we would like to mention is that SCALE's novelty lies in joint perception-action modulation in VLAs that is not explored in prior arts, not in what components we used. By the joint modulation, we observe (Table 6 in the submission) that adaptive decoding and adaptive visual attention each contribute complementary gains (+5.3% and +3.3%, respectively), yet their combination yields +**10.6**%—exceeding the sum of individual gains (+8.6%) by over **2**%. This shows not only complementariness of each component's contribution but also a synergistic effect. It implies a coupled interaction where improved perception produces better uncertainty estimates that in turn guide action decoding, and vice versa.
> > >
> > > As discussed in Sec. 1 and 3, this coupled perception-action loop is conceptually motivated by Active Inference theory [1], which suggests that agents benefit from jointly adapting perception and action under uncertainty—a principle that SCALE operationalizes for VLAs in a practical, training-free manner.
> > >
> > > Again, we appreciate your comments and cordially ask you to take a second look at your argument.
> > >
> > > ---
> > >
> > > [1] Friston et al., Active inference and learning. Neuroscience & Biobehavioral Reviews, 2016.

---

### Official Review · Reviewer_PHyX · 2026-03-13

**Soundness:** 3
**Presentation:** 2
**Significance:** 3
**Originality:** 3
**Overall Recommendation:** 5
**Confidence:** 4

**Summary:**

This work proposes an uncertainty-aware inference strategy for autoregressive VLAs named SCALE, where the core idea is to modulate both the visual attention and action prediction using "self-uncertainty", which leverages the log-likelihood ratio between the model's logit output distribution and two other states: uniform distribution (high uncertainty) and a one-hot-like distribution (low uncertainty).
Under low uncertainty, SCALE sharpens visual attention and executes actions nearly greedily; under high uncertainty, it broadens visual attention to explore the scene and increases action sampling diversity. SCALE is training-free, requires no external verifiers, and operates in a single forward pass per control step.

**Compliance With Llm Reviewing Policy:**

Affirmed.

**Final Justification:**

The rebuttal has resolved my questions and concerns. My score is kept (see rebuttal acknowledgment).

**Key Questions For Authors:**

1. Could the authors clarify/discuss the connection to the terms in the expression/equations for the expected free energy?
2. Can/Could the proposed self-uncertainty distinguish between epistemic and aleatoric uncertainty contributions?
3. Figure 5 shows that lower max probability correlates with lower success rates. Are there instances in the OOD real-world tasks where the model was highly confident but incorrect? If that is the case, how did SCALE behave in those specific failure modes?

**Limitations:**

The authors adequately address the limitations (reliance on previous-step uncertainty) in the appendices and provide strong empirical justification for their design choices. However, the authors do not discuss:
- SCALE's self-uncertainty capacity to distinguish between epistemic and aleatoric uncertainty (uncertainty disentanglement).
- The restriction to autoregressive VLAs, excluding diffusion-based approaches.

**Strengths And Weaknesses:**

---
### Strengths

- The paper is well written, and the figures and illustrations are well placed, helping to digest the paper's information and the proposed method.
- The single-pass, training-free design is highly relevant, as most uncertainty-aware methods add significant computational overhead, making the proposed method particularly appealing to practitioners.
- The comparison of the double logit distribution with certain and uncertain outcomes represents an interesting and novel approach for estimating self-uncertainty.
- Compelling results across extensive validation of the method in simulated and real-world experiments, backed by well-designed ablation studies.

---
### Weaknesses

- **The Active Inference connection is conceptual rather than formal, and the framing may mislead readers.** The paper invokes Active Inference as "theoretical grounding" for its explore-exploit mechanism, but never establishes a formal link. No derivation connects the self-uncertainty measure to any term in the variational free energy and expected free energy equations. The language escalates across the paper from "inspired by" (Abstract, L026-027) to invoking specific concepts like expected free energy and information gain (Section 3, L141-144, right col.) to claiming the framework "provides theoretical grounding" (Appendix A, L621-622), creating the impression that SCALE is derived from or justified by the Active Inference formalism. The authors should either develop the formal link or reframe Active Inference as a conceptual motivation rather than a theoretical foundation.

- Relying on logits distribution remains a single point of failure, having a miss-calibrated prediction could trigger the opposite of the wanted behavior from the method.

- The reliance on the uncertainty of the previous step has been addressed by the authors, but the effectiveness of the method in quick action-response use-cases remains unknown. Readers must be informed of the potential ineffectiveness of this method in such scenarios in the main paper.

---

> ### Author Rebuttal · Authors · 2026-03-31
>
> > (**W1, Q1**): The paper presents Active Inference as "theoretical grounding," but the connection remains conceptual—no formal derivation links the self-uncertainty measure to any term in the variational or expected free energy equations. The authors should either formalize this link or reframe Active Inference as conceptual motivation rather than a theoretical foundation.
>
> $\to$ We intended Active Inference as a conceptual motivation, not a theoretical foundation; we will make this distinction explicit in the revision. Both EFE [1] and our $u^k_t$ balance two competing terms to govern the explore--exploit trade-off, and SCALE operationalizes this principle in a training-free manner for VLA control, with the measure itself grounded in log-likelihood ratio testing [2]. Structurally, $u^k_t = D_{\mathrm{KL}}(p^k_t \| q^{\mathrm{low}}) - D_{\mathrm{KL}}(p^k_t \| q^{\mathrm{high}})$ parallels EFE's pragmatic--epistemic decomposition, but the individual terms differ: EFE's concern preferred outcomes and information gain [1], whereas ours quantify distances from full certainty and full ambiguity. We will clarify this in the revised manuscript.
>
> [1] Friston et al., Active inference..., Neurosci. Biobehav. Rev., 2016.\
> [2] Neyman & Pearson, On the problem…, Philosophical Transactions of the Royal Society, 1933.
>
> ---
>
> > (**W2,Q3**): Relying solely on the logit distribution is a potential single point of failure—miscalibrated confidence could trigger the opposite of the intended behavior. Are there OOD real-world cases where the model was highly confident but incorrect, and how did SCALE behave in those failure modes?
>
> $\to$ Within our experimental settings, the logit distribution carries task-relevant information: episodes with low $p_{\max}$ exhibit lower success rates (Figure 5), and SCALE consistently improves over greedy decoding across in-domain benchmarks (Tables 1-3). Confident-but-incorrect instances can indeed exist in OOD tasks, as overconfidence is a known challenge for neural networks [1], including LLMs [2] and VLAs [3]. In such cases, SCALE's $u^k_t$ remains low, but unlike greedy decoding, SCALE's non-zero sampling temperature still allows exploration beyond the top-1 token. This is reflected in consistent gains in both unseen simulation (Table 4; OpenVLA: 18.0\%$\to$21.5\%, $\pi_0$-FAST: 35.7\%$\to$38.8\%) and OOD real-world experiments (Table 5; OpenVLA: 22.9\%$\to$39.6\%, $\pi_0$-FAST: 43.8\%$\to$56.3\%). Nonetheless, integrating calibration methods compatible with SCALE's training-free, single-inference design is an interesting direction for future work.
>
> [1] Guo et al., On Calibration of Modern…, ICML, 2017.\
> [2] Geng et al., A Survey of Confidence…, NAACL, 2024.\
> [3] Zollo & Zemel, Confidence Calibration…, arXiv, 2025.
>
> ---
>
> > (**W3**): How does SCALE perform in quick action-response scenarios, and could the authors explicitly discuss this limitation in the main paper?
>
> $\to$ We conducted additional dynamic experiments using $\pi_0$-FAST [1]. In simulation (DOMINO benchmark [2], Table A), SCALE improves over greedy decoding (19.8\%$\to$26.5\%), with the two-step oracle achieving further gains (29.5\%). In real-world OOD dynamic pick tasks (5cm/s, Table B), SCALE again improves (52\%$\to$68\%), and the two-step oracle reaches 72\%. However, at higher speed (10cm/s), all methods achieve zero success---a limitation inherited from the base VLA architecture, as noted by [1]. We will discuss this trade-off in the revised manuscript.
>
> | Method | shake bottle | shake bottle horizontally | adjust bottle | place container plate | Avg. |
> | :--- | :---: | :---: | :---: | :---: | :---: |
> | $\pi_0$-FAST (greedy) | 27 | 28 | 11 | 13 | 19.8 |
> | + SCALE (**ours**) | 35 | 35 | 19 | 17 | 26.5 |
> | + Two-Step Oracle | 38 | 37 | 21 | 22 | 29.5 |
>
> **Table A.** Success rates (%) on dynamic DOMINO [2] simulation tasks (100 episodes per task).
>
> | Method | Moving object (5 cm/s) | Moving object (10 cm/s) |
> | :--- | :---: | :---: |
> | $\pi_0$-FAST (greedy) | 52 | 0 |
> |  + SCALE (**ours**) | 68 | 0 |
> |  + Two-Step Oracle | 72 | 0 |
>
> **Table B.** Success rates (%) on a real-world dynamic task (“Pick up the lemon”) at varying speeds (25 episodes for each speed).
>
> [1] Pertsch et al., Fast: Efficient..., RSS, 2025.\
> [2] Fang et al., Towards Generalizable... arXiv, 2026.
>
> ---
>
> > (**Q2**): Can/Could the proposed self-uncertainty distinguish between epistemic and aleatoric uncertainty contributions?
>
> $\to$ It cannot. Both epistemic and aleatoric uncertainty jointly shape $p^k_t$, and disentangling them from a single forward pass is inherently difficult [1]. SCALE instead treats uncertainty as a unified signal for behavioral modulation (explore *vs.* exploit), empirically validated across diverse benchmarks (Tables 1--5). Disentangling the two while preserving single-pass efficiency is a valuable future direction.
>
> [1] Kendall & Gal, What Uncertainties…, NeurIPS, 2017.

---

> > ### Author Rebuttal · Reviewer_PHyX · 2026-04-01
> >
> > The authors have addressed my concerns and questions. Based on the additional results and the clarifications for the revised manuscript, I decided to increase my score from 4 to 5.

---

> > > ### Author Response · Authors · 2026-04-02
> > >
> > > We thank Reviewer **PHyX** for the encouraging feedback. We are pleased that our additional results and clarifications addressed your concerns and questions. We sincerely appreciate your updated score.
> > >
> > > If you have any further questions about the paper, please feel free to ask; we would be happy to respond.
> > >
> > > Best regards,\
> > > Authors

---

### Decision · Program_Chairs · 2026-04-30

**Decision:**

Accept (spotlight)

**Comment:**

This paper proposes SCALE, a training‑free single‑pass uncertainty method to improve VLA robustness. Reviewers praise its elegance, practical design, and extensive experiments across backbones. However, weaknesses limit the contribution: the Active Inference framing is conceptual, not formal; novelty is limited (standard techniques); hyperparameters are sensitive and model‑dependent; real‑world tasks are too simple (static pick‑and‑place). Most of these issues are addressed in the rebuttal.

Overall, the paper shows interesting new ideas that are worth exploring.